# Novel repertoire of tau biosensors to monitor pathological tau transformation and seeding activity in living cells

Erika Cecon[1], Atsuro Oishi[1], Marine Luka[1], Delphine Ndiaye-Lobry[1], Arnaud François[2], Mathias Lescuyer[1], Fany Panayi[2], Julie Dam[1], Patricia Machado[2], Ralf Jockers[1]*

[1]Institut Cochin, Inserm U1016, CNRS UMR 8104, Université de Paris, Paris, France; [2]Les Laboratoires Servier, Suresnes, France

**Abstract** Aggregates of the tau protein are a well-known hallmark of several neurodegenerative diseases, collectively referred to as tauopathies, including frontal temporal dementia and Alzheimer's disease (AD). Monitoring the transformation process of tau from physiological monomers into pathological oligomers or aggregates in a high-throughput, quantitative manner and in a cellular context is still a major challenge in the field. Identifying molecules able to interfere with those processes is of high therapeutic interest. Here, we developed a series of inter- and intramolecular tau biosensors based on the highly sensitive Nanoluciferase (Nluc) binary technology (NanoBiT) able to monitor the pathological conformational change and self-interaction of tau in living cells. Our repertoire of tau biosensors reliably reports *i.* molecular proximity of physiological full-length tau at microtubules; *ii.* changes in tau conformation and self-interaction associated with tau phosphorylation, as well as *iii.* tau interaction induced by seeds of recombinant tau or from mouse brain lysates of a mouse model of tau pathology. By comparing biosensors comprising different tau forms (*i.e.* full-length or short fragments, wild-type, or the disease-associated tau(P301L) variant) further insights into the tau transformation process are obtained. Proof-of-concept data for the high-throughput suitability and identification of molecules interfering with the pathological tau transformation processes are presented. This novel repertoire of tau biosensors is aimed to boost the disclosure of molecular mechanisms underlying pathological tau transformation in living cells and to discover new drug candidates for tau-related neurodegenerative diseases.

## Editor's evaluation

Understanding specific tau-tau interactions that play key roles in Alzheimer's disease and tauopathies will enable the elucidation of the toxic tau species involved in the pathogenesis of these diseases and therapeutic development in this area. In their important paper, Cecon et al. develop a series of NanoBiT complementation-based tau biosensors to monitor tau intramolecular and intermolecular interactions. This solid work will be of high interest to a broad target audience including researchers in the field of biophysics, biochemistry, cell biology, neuroscience, neuropathology, and drug discovery.

## Introduction

Tauopathies are a group of neurodegenerative diseases that display abnormal neuronal aggregates of the tau protein, also called neurofibrillary tangles (NFT). Examples of tauopathies include AD, the most common dementia worldwide, frontotemporal dementia (FTD), Pick disease, and others. Under

*For correspondence: ralf.jockers@inserm.fr

physiological conditions, tau is found mainly associated with microtubules (MTs), participating in the regulation of MT assembly/disassembly, with an impact on axonal transport and synapse structural organization. In AD and other tauopathies, tau undergoes a pathological transformation that involves hyperphosphorylation, conformational changes, loss of MT-binding affinity, and oligomerization, ultimately resulting in tau mislocation and NFTs formation (*Ugalde et al., 2016*; *Goedert, 2015*). Progressive accumulation of NFTs in specific brain areas is the molecular marker that mostly correlates with AD cognitive dysfunction and is the basis of the Braak classification of AD into VI stages (*Braak and Braak, 1996*).

Tau is encoded by the *MAPT* gene and is expressed as six different isoforms due to alternative splicing, referred to as 0N3R, 1N3R, 2N3R, 0N4R, 1N4R, and 2N4R, depending on the presence or absence of two near-amino-terminal inserts (N) and of three (3 R) or four (4 R) repeats in its central repeat domain (*Mandelkow and Mandelkow, 2012*). Except for the 0N3R isoform, which is found only in newborns, all the other isoforms are expressed in the human brain and all can be found in aggregates observed in AD brains. Tau phosphorylation/dephosphorylation determines its affinity to bind to MTs and this is a tightly controlled and dynamic process under physiological conditions. Disease-associated tau is hyperphosphorylated and shows the lower capacity to bind MTs while its propensity to aggregate is increased. Once started, tau pathology is propagated and spread between neurons in a prion-like mechanism, transmitting the misfolded pathological conformation to other native tau molecules through a prion-like seeding effect (*Iba et al., 2013*). Whilst tau conformational changes, self-interaction, and aggregation are clearly linked to tauopathies, the underlying molecular mechanisms are still poorly understood, limiting thus the development of therapeutics targeting these processes.

Tau self-interaction has been extensively studied using in vitro aggregation reporter systems like the fluorescent reporters thioflavin T or congo red. Those experiments provided important information on the kinetics of fibrilformation and on the in vitro conditions to induce tau aggregation, such as the requirement of heparin or other polyanions (*Goedert et al., 1996*). However, these assays rely on high concentrations (μM to mM) of tau and, thus, are unlikely to fully mimic the transformation process that occurs in a cellular context. The development of cell-based sensors overcame part of these obstacles. Initial assays were mainly based on visualization of intracellular tau aggregates by western blot or fluorescent microscopy using antibodies or recombinant tau molecules fused to fluorescent proteins (GFP, CFP, YFP), or complementation assays based on the reconstitution of fluorescent proteins (BiFC) (*Chun and Johnson, 2007*; *Nonaka et al., 2010*; *Guo and Lee, 2011*; *Xu et al., 2016*; *Chun et al., 2007*; *Tak et al., 2013*; *Lim et al., 2015*). A Fluorescence Resonance Energy Transfer (FRET)-based biosensor, combined with microscopy imaging or fluorescence-assisted cell sorting (FACS), has been widely used for the detection of tau seeding activity and propagation (*Holmes et al., 2014*; *Kfoury et al., 2012*). More recently, FRET-based biosensors combined with fluorescence lifetime (FLT) detection allowed improved quantification of the signal for screening applications (*Lo et al., 2019*). Inspired by these seminal works, we sought to develop novel tau biosensors displaying high-throughput screening compatibility (*Lo, 2021*) and high sensitivity, based on the recently developed Nluc NanoBiT (*Dixon et al., 2016*; *Hall et al., 2012*; *Cooley et al., 2020*). Nluc assays are highly sensitive, have a large dynamic range and are based on a simple and quantitative readout, the generation of luminescence light. The size of the Nluc (19 kDa) is small compared to typical fluorescent proteins minimizing the impact of the fusion protein on the function of the target protein. This technology has been previously applied to investigate the effect of kinase inhibitors on spontaneous tau self-aggregation (*Holzer et al., 2018*).

Here, we have developed a series of tau biosensors using the NanoBiT complementation system to monitor spontaneous and phosphorylation- and seeding-induced *intra*molecular conformational changes, as well as *inter*molecular tau self-interaction of a variety of tau forms, in living cells and in a quantitative and high-throughput-compatible manner.

## Results

### Development of Nanoluciferase enzyme complementation-based tau biosensors

In order to monitor changes in tau conformation and aggregation we developed intra- and intermolecular biosensors, respectively, using the NanoBiT technology. In this system, the Nluc enzyme is split into two fragments, a large 18 kDa fragment (LgBit) and a smaller 1.3 kDa fragment (SmBit), that are fused to the proteins of interest (*Dixon et al., 2016*). To design tau biosensors using this system, the LgBit was fused to the N-terminus of tau, and the SmBit either at the C-terminus of the double-fused tau (*intra*molecular sensors) or to the N-terminus of single-fused tau (*inter*molecular sensor) (*Figure 1A*). Different forms of tau exist. They contain or not have two domains at the N-terminal part (0 N-2N) and vary in the number of repeats (R1 to R4) at their central repeat domain. In this study we used full-length (2N4R) wild-type tau (WT-Tau), full-length tau carrying the pro-aggregating P301L mutation (Tau(P301L)), and the pro-aggregating K18 fragment of tau (*Li and Lee, 2006*; *Michel et al., 2014*; *Shammas et al., 2015*) comprising only the four-repeat domains (R1-R4) and carrying the P301L mutation (K18(P301L)) (*Figure 1B*). Among the advantages of the NanoBiT system used for these tau biosensors is that it allows monitoring the behavior of tau molecules in real-time, in living cells, in a straightforward pipeline. As shown in *Figure 1C*, cells can be treated with test compounds, and Nluc activity is measured immediately after the addition of its substrate, reflecting the fast and high-throughput properties of the assay.

In order to validate the NanoBiT system for the use in tau biosensors, we first developed an intermolecular tau interaction biosensor comprising the aggregation-prone K18 fragment displaying the P301L mutation (K18(P301L)), as the aggregation properties of this tau fragment have been extensively reported (*Li and Lee, 2006*; *Michel et al., 2014*; *Shammas et al., 2015*). HEK293T cells expressing a fixed amount of LgBit-K18(P301L) and increasing amounts of SmBit-K18(P301L), showed the expected saturation profile of the complementation signal, reflecting specific interaction and progressive saturation of LgBit-K18(P301L) with SmBit-K18(P301L) (*Figure 1D*). Expression levels of the sensor were monitored in parallel by western-blot using anti-HA antibodies recognizing both constructs (*Figure 1E*). The cellular distribution of the K18(P301L) biosensor was visualized by immunofluorescence microscopy using an antibody against the HA tag of the biosensor. The K18(P301L) biosensor showed a punctuated staining pattern predominantly in the cytosol with some co-localization with tubulin as well as some nuclear staining (*Figure 1F*). Next, we challenged biosensor cells with recombinant K18 protein, a treatment that has been previously reported to efficiently induce ('seed') tau aggregation (*Holmes et al., 2014*; *Frost et al., 2009*). Heparin-induced aggregates of recombinant K18 (aggK18) and of full-length tau (aggTau) were obtained as previously reported (*Mirbaha et al., 2015*). A robust increase in signal was observed in K18(P301L) sensor-expressing cells treated for 24 hr with either monomeric K18 (mK18) or aggK18 (*Figure 1G*). Seeds of aggTau also led to a significant increase in the sensor signal, while monomeric tau (mTau) and oligomers of the amyloid-beta peptide (oAß) had no effect (*Figure 1G*). No further increase in sensor signal was observed up to 48 hr of seed treatment (not shown). The presence of ß-sheet oligomers of Aß and of tau and K18 aggregates in the respective preparations has been confirmed by thioflavin T (ThT) assay (*Figure 1—figure supplement 1A*). Altogether, these results confirm the specificity of the sensor response towards tau seeds and the lack of cross-seeding activity to Aß. Concentration-response experiments of aggK18 show that the biosensor is sensitive to subnanomolar concentrations of seeds, displaying a twofold increase in signal at 0.3 nM of aggK18 (*Figure 1H*). No aggK18-induced increase in the complementation signal was observed when either LgBit-K18(P301L) or SmBit-K18(P301L) were expressed alone (*Figure 1—figure supplement 1B–C*). Of note, the kinetics of the Nluc enzyme:substrate reaction reaches a plateau in less than 10 min after substrate addition (*Figure 1—figure supplement 1D*), and all results are taken at this time-point to avoid any interference of the enzymatic activity kinetics on the comparison between different treatments. These results demonstrate the suitability of the NanoBiT technology to monitor the pathologically relevant seeding response and self-interaction of the K18(P301L) tau fragment.

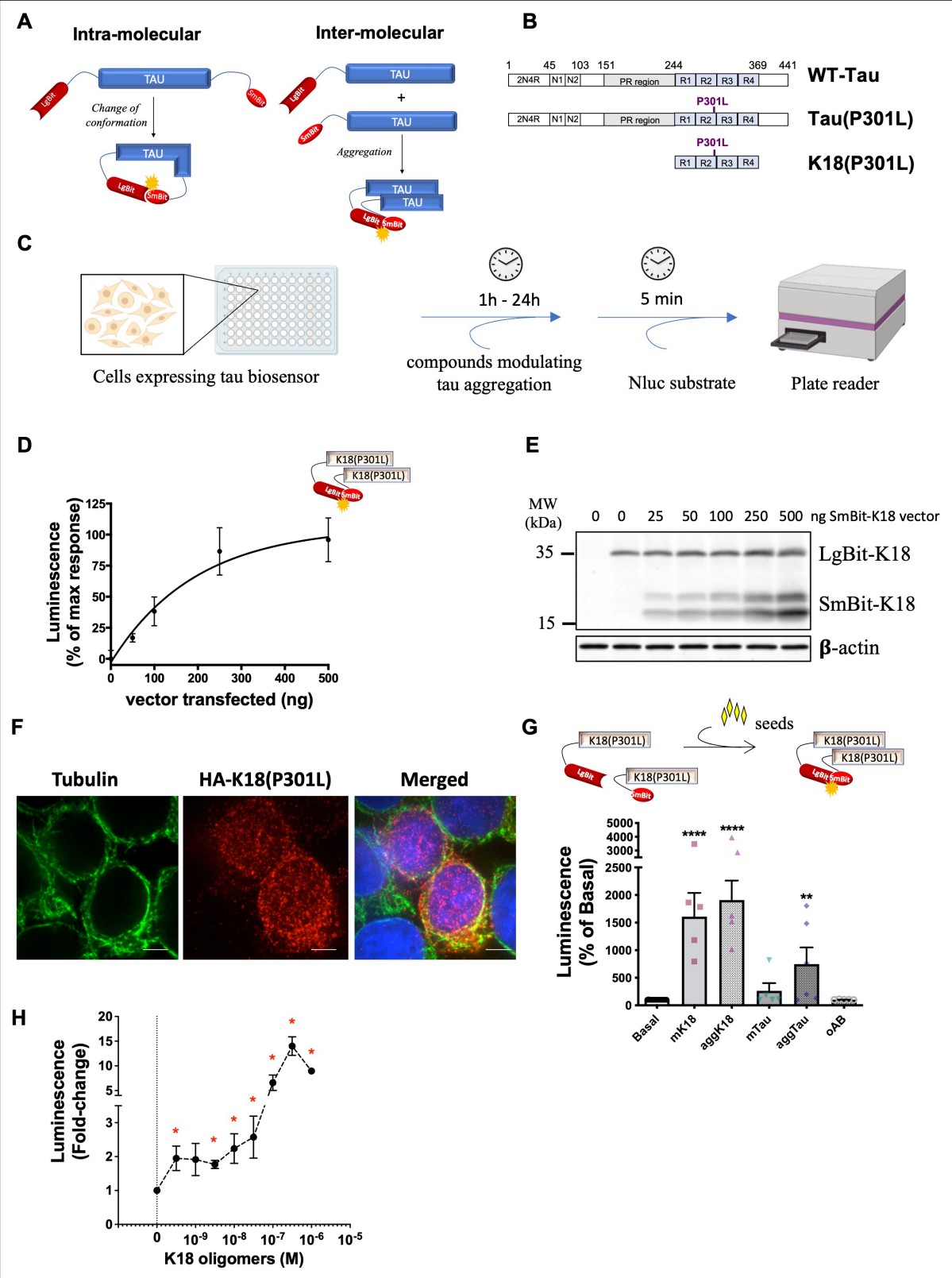

**Figure 1.** NanoBiT-based biosensors development and application to monitor the tau seeding response in living cells. (**A**) Scheme of NanoBiT-based intramolecular (conformational) and intermolecular (interaction) biosensors. (**B**) Biosensors were developed for three tau forms: full-length wild-type tau (WT-Tau) corresponding to the 2N4R isoform displaying both of the two domains at the N-terminal part (N1, N2), the proline-rich (PR) region, and four-repeat domains (R1–R4); full-length tau carrying the P301L mutation (Tau(P301L)); and the tau fragment comprising only the four repeat domains

*Figure 1 continued on next page*

*Figure 1 continued*
(R1–R4) and carrying the P301L mutation (K18(P301L)). (**C**) Workflow of NanoBiT assay (scheme created with Biorender). (**D**) Saturation curve of the intermolecular interaction sensor K18(P301L) in HEK293T cells expressing a fixed amount of LgBit-K18(P301L) (250 ng of a transfected vector) and increasing amounts of SmBit-K18(P301L) (0–500 ng of a transfected vector); data are expressed as mean ± SEM of four independent experiments. (**E**) Representative western blot analysis of the expression of K18(P301L) biosensor pairs used in panel D. (**F**) Cellular distribution of the K18(P301L) biosensor monitored by immunofluorescence microscopy (tubulin: in green; biosensor: anti-HA antibody, in red; nuclei: DAPI staining, in blue). Scale bar: 10 µm. (**G**) K18(P301S) interaction biosensor signal in the presence of recombinant monomeric K18 (mK18; 100 nM), aggregated K18 (aggK18; 100 nM) or tau (aggTau; 100 nM), or oligomeric amyloid beta peptide (oAb; 1 µM); data are expressed as mean ± SEM of five independent experiments; **p<0.01, ****p<0.001 by one-ANOVA, followed by Dunnett's multiple comparisons test to basal condition. (**H**) K18(P301S) interaction biosensor signal in the presence of increasing concentrations of recombinant oligomeric K18 (0.3 nM to 300 nM); data are expressed as mean ± SEM of four independent experiments; * p<0.05 compared to control (0) by Student *t*-test.

The online version of this article includes the following source data and figure supplement(s) for figure 1:

**Source data 1.** Unedited blot of *Figure 1E*.

**Figure supplement 1.** Validation of recombinant proteins and the NanoBiT-based K18(P301S) interaction sensor.

## Tau biosensors report conformational changes and tau-tau interaction in response to microtubule destabilization and phosphorylation

The conditions triggering the transformation processes of full-length wild-type tau in a cellular context are largely unknown. We, therefore, aimed to apply the NanoBiT approach to full-length wild-type tau. The WT-Tau intermolecular interaction sensor (LgBit-Tau/SmBit-Tau) behaved similarly to the K18(P301L) interaction sensor showing a robust saturation curve in HEK293T cells expressing a fixed amount of LgBit-Tau and increasing amounts of SmBit-Tau (*Figure 2A*). Replacement of the SmBit-Tau by the non-related Halo protein fused to SmBit generated no signal confirming the specificity of the tau intermolecular self-interaction assay (*Figure 2A*). Successful expression of sensors at the protein level was monitored by western-blot using the HA tag common to all constructs (*Figure 2A*, **bottom**).

The central repeat domain of tau is known to be involved in MT binding and in the pathological intermolecular interaction of tau (*Mukrasch et al., 2005*), meaning that both events are mutually exclusive. To determine whether the basal signal observed with the WT-Tau interaction sensor originates from MT-bound tau or cytosolic tau, we treated biosensor cells with colchicine to destabilize MTs. Under this condition, the sensor signal was decreased by 75% (*Figure 2B*) indicating that the basal signal most likely reflects molecular proximity between MT-bound tau molecules rather than self-interaction of cytoplasmic tau. Consistently, colchicine had no effect on the basal signal of the K18(P301L) interaction sensor, suggesting that the signal of this sensor is MT-independent and reflects K18(P301L) self-interaction in the cytosol (*Figure 2C*). A potential direct impact of colchicine on preventing tau-tau interactions was discarded by monitoring in vitro tau aggregation (induced by heparin) in the ThT assay (*Figure 2—figure supplement 1A*). Immunofluorescence microscopy revealed a strong colocalization of the tau sensor with MTs (*Figure 2D* **top panel**), which corresponds to the reported localization of endogenous full-length tau and is distinct from the predominant cytoplasmic localization of K18 (*Frost et al., 2009*). Instead, colchicine-treated cells showed disorganized MTs, as expected, and largely diminished colocalization between the tau sensor and tubulin staining (*Figure 2D* **bottom panel**). Collectively, these data indicate that the intermolecular tau sensor, like endogenous tau, is located at the MT and generates a complementation signal reflecting molecular proximity between tau molecules that are attached to MTs.

We then probed the conformational state of full-length tau with the *intra*molecular tau conformational biosensor where LgBit and SmBit are fused to the same tau molecule (LgBit-Tau-SmBit). Expression of increasing quantities of the sensor resulted in a linear increase of the luminescence signal as would be expected for an intramolecular complementation signal (*Figure 2E*). Of note, under equal expression levels, the basal raw luminescence signal of the *intra*molecular sensor is much higher (approximately 10 times) than that of the *inter*molecular sensor (*Figure 2F*). This data indicates close proximity between the N- and C-terminal domains of tau, which is consistent with previous in vitro observations and molecular models showing that MTs stabilize tau into a 'paperclip fold' structure, in contrast to the natively unfolded structure of soluble tau (*Mandelkow et al., 2007*; *Mukrasch et al., 2009*; *Di Primio et al., 2017*). Our data thus suggests that the large majority of the signal comes indeed from the intramolecular LgBit-SmBit complementation, a conclusion that is in agreement with the previous observations using a FRET-based intramolecular sensor (*Di Primio et al., 2017*).

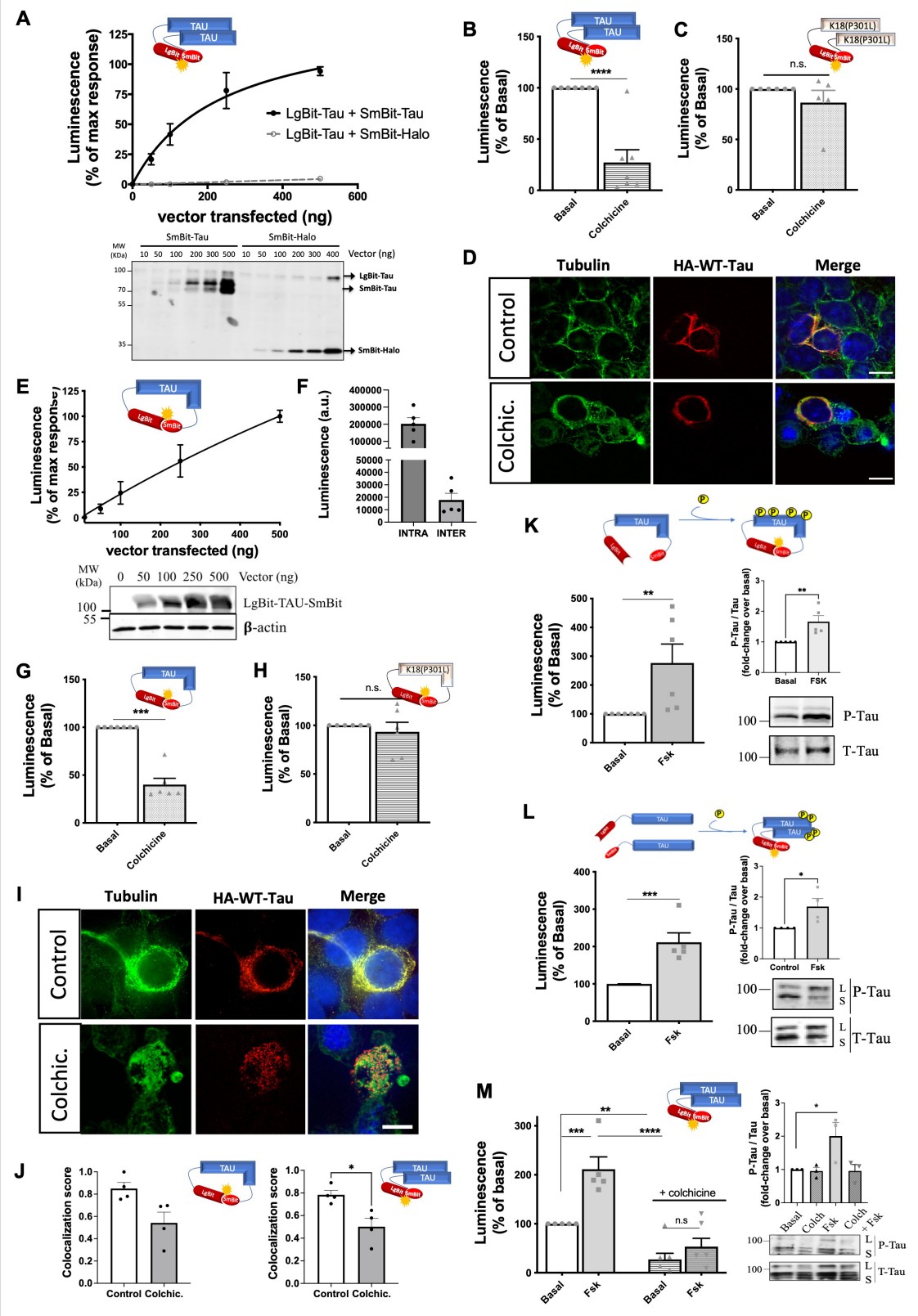

**Figure 2.** Tau biosensors monitor conformational change and self-interaction of wild-type (WT) full-length tau. (**A**) Saturation curve of the intermolecular WT-Tau interaction sensor in HEK293T cells expressing a fixed amount of LgBit-Tau (250 ng of a transfected vector) and increasing amounts of SmBit-Tau or SmBit-Halo (0–500 ng of a transfected vector); data are expressed as mean ± SEM of five independent experiments. Bottom: Representative western blot (WB) analysis of the expression of the tau biosensor pairs. (**B–C**) Signals of WT-Tau (**B**) and K18(P301L) (**C**) interaction biosensors in cells treated or

*Figure 2 continued*

not with colchicine (10 µM, 1 hr); data are expressed as mean ± SEM of seven and five independent experiments, respectively; **p<0.005 by two-tailed paired Student *t*-test; n.s.=not-significant. (**D**) Cellular distribution of the WT-Tau interaction biosensor monitored by immunofluorescence microscopy in the absence (top panels) or presence (bottom panels) of colchicine (biosensor: anti-HA antibody, in red; tubulin: in green; nuclei: DAPI staining, in blue); Scale bar: 10 µm. (**E**) WT-Tau conformational sensor signal in HEK293T cells expressing increasing amounts of LgBit-Tau-SmBit vector (0–500 ng of a transfected vector); data are expressed as mean ± SEM of seven independent experiments. Bottom: Representative WB analysis of the expression of the Tau conformational biosensor. (**F**) Basal luminescence signal of the WT-Tau conformational (INTRA) and WT-Tau interaction (INTER) biosensors at the same expression levels; data are expressed as mean ± SEM of five independent experiments. (**G–H**) WT-Tau (**G**) and K18(P301L) (**H**) conformational biosensors signal in cells treated or not with colchicine; data are expressed as mean ± SEM of seven (**G**) or five (**H**) independent experiments; ***p<0.001 by two-tailed paired Student *t*-test; n.s.=not-significant. (**I**) Cellular distribution of the WT-Tau conformational biosensor monitored by immunofluorescence microscopy in the absence (top panels) or presence (bottom panels) of colchicine (biosensor: anti-HA antibody, in red; tubulin: in green; nuclei: DAPI staining, in blue); Scale bar: 10 µm. (**J**) Colocalization analysis (Manders' colocalization coefficient) of the fractional overlap between biosensor molecules and MTs. (**K–L**) Signals of WT-Tau conformational (**K**) and interaction (**L**) biosensors in cells treated or not with forskolin (10 µM, 24 hr); data are expressed as mean ± SEM of six independent experiments; **p<0.01, ***p<0.005 by two-tailed paired Student *t*-test; right panel: WB analysis and quantification of phosphorylated tau (P-Tau) and Total tau (T-tau) using AT8 and anti-HA antibodies, respectively. L: Lgbit-Tau; S: SmBit-Tau. (**M**) WT-Tau interaction biosensor response in cells treated with colchicine (10 µM, 1 hr), forskolin (10 µM, 24 hr), or both (colchicine followed by forskolin); data are expressed as mean ± SEM of five independent experiments; ** p<0.01, ***p<0.005, ****p<0.001 by one-way ANOVA followed by Sidak's multiple comparison test.

The online version of this article includes the following source data and figure supplement(s) for figure 2:

**Source data 1.** Unedited blot of *Figure 2A*.

**Source data 2.** Unedited blot of *Figure 2E*.

**Source data 3.** Unedited blot of *Figure 2K*.

**Source data 4.** Unedited blot of *Figure 2L*.

**Source data 5.** Unedited blot of *Figure 2M*.

**Figure supplement 1.** Effect of microtubules (MT) destabilization and phosphorylation on WT-Tau biosensors.

Treatment of WT tau sensor cells with colchicine reduced the signal of the tau conformation sensor by 60%, consistent with the idea that soluble tau switches to a different structure when detaching from MTs (*Figure 2G*), whereas the signal of a LgBit-K18(P301L)-SmBit conformation sensor was unaffected as expected (*Figure 2H*). Immunofluorescence microscopy confirmed the localization of the LgBit-Tau-SmBit sensor at MTs, while colchicine treatment partially disrupted it (*Figure 2I*). Colocalization analysis shows that the cell distribution of the majority of tau sensor molecules of both intra- and inter-molecular WT tau sensors overlap with MT staining (Manders' Coefficient score of 0.85 and 0.78, respectively), and this colocalization is significantly decreased in colchicine-treated cells (score of 0.54 and 0.5, respectively; *Figure 2J*). Treatment with nocodazole, which also disturbs MT organization, led to a decrease in the Nluc complementation of both WT-tau sensors (inter- and intramolecular) and no change in the K18 sensor signal, similar, to what was observed with colchicine (*Figure 2—figure supplement 1B–D*). Taken together, these data indicate that the full-length tau conformation sensor faithfully monitors different conformational states of tau depending on its MTs-bound or unbound status.

In the course of its pathological transformation, tau is known to become hyperphosphorylated and to dissociate from MTs (*Brandt et al., 2020*; *Xia et al., 2021*). However, the link of these events to tau transformation (conformational changes, seeding, aggregation) is not clear. To address this question, we induced tau phosphorylation by treating biosensor cells expressing either the conformation or the interaction WT-Tau sensors with forskolin for 24 hr. Forskolin directly activates the enzyme adenylyl cyclase, increasing the intracellular cAMP levels, which then leads to the activation of the cAMP-dependent kinase PKA that can phosphorylate tau (*Seamon and Daly, 1986*). For both biosensors, a 2–3-fold increase in signal was observed in the presence of forskolin, indicative of the induction of a change in conformation (*Figure 2K*) and the induction of tau self-interaction (*Figure 2L*). Phosphorylation of tau biosensors induced by forskolin treatment was confirmed by the increased immunoreactivity of the anti-phospho-tau specific AT8 antibody (Ser202/Thr205), a phosphorylation site that correlates well with pathological tau species observed in AD (*Zhou et al., 2006*; *Figure 2K–L* insets). In order to assess whether phosphorylation-induced self-interaction of WT tau biosensor can be favored by increasing the MT-free population of tau, we treated sensor cells with colchicine followed by forskolin. Surprisingly, the co-treatment condition completely abrogated the forskolin effect on the

sensor signal (*Figure 2M*) as well as on tau phosphorylation (*Figure 2M* **inset**), suggesting that WT-tau phosphorylation occurs mainly at MT-bound tau. Accordingly, the WT-tau intramolecular conformational sensor followed a similar pattern, further supporting that the phosphorylation-induced conformational change is abrogated in MT-unbound tau (*Figure 2—figure supplement 1E*). An alternative method to favor tau phosphorylation is by inhibiting phosphatase activity with the use of okadaic acid (OA). Treatment of cells with OA for 24 hr resulted in an increase in the interaction WT-Tau sensor signal, similar to forskolin treatment, which is also lost by colchicine (*Figure 2—figure supplement 1F*). The effect of forskolin on the WT-Tau conformation sensor signal was recapitulated in the SH-SY-5Y neuronal cell line, although with a lower amplitude compared to HEK293 cells probably due to a lower number of transfected cells. Expression of the sensor in SH-SY-5Y cells was validated by immunofluorescence microscopy and its cell distribution follows MTs distribution pattern (*Figure 2—figure supplement 1G*). These data, together with the fact that HEK cells are not only cells where the biosensors are responsive like in neurons, but also easy to manipulate, led us to favor the use of HEK cells for the expression of the biosensors. In summary, our tau biosensors have proven to be useful tools to monitor the behavior of tau in a physiological cellular context, while enabling the investigation of different factors suspected to promote changes in its conformation and self-interaction properties.

## Seeding response of tau biosensors

The mechanisms underlying the aggregation of physiological forms of tau remains poorly known. To investigate whether the high sensitivity of the NanoBiT system is able to report on the seeding response of physiological tau we assessed the response of our WT-Tau interaction sensor towards different seeds. Differently from the K18(P301L) interaction sensor (see *Figure 1G–H*), none of the recombinant K18 or tau seeds induced self-interaction of the WT-Tau biosensor when run in parallel (*Figure 3A*). The WT-Tau interaction biosensor also did not respond to seeds obtained from brain lysates of the transgenic (Tg) tauopathy mouse model carrying the P301S mutation (*Figure 3B*), while the K18(P301L) interaction biosensor cells showed a robust 107-fold increase in signal in the presence of brain lysates from the Tg mice compared to brain lysates from WT mice (*Figure 3B*). Such an increase was not observed in biosensor cells expressing LgBit-K18(P301L) and the non-related SmBit-Halo protein (*Figure 3—figure supplement 1A*), confirming the specificity of the biosensor response. The K18(P301L) seeding response to Tg brain lysates was concentration-dependent, reaching saturation at around 1 µg of protein lysate (*Figure 3C*). This sensor showed a large dynamic range spanning three logs of brain lysate concentrations (*Figure 3C*), with a minimum detection limit of 300 pg of total protein (*Figure 3C* **inset**). The robustness and reproducibility of this biosensor response were further confirmed using brain lysates from four independent WT and Tg mice to account for interindividual variability in tauopathy development (*Figure 3—figure supplement 1B*). This biosensor showed robust discrimination between WT and Tg brain lysates also in a 384-well format (luminescence signal: 932 vs 23,580 units.; *Figure 3—figure supplement 1C*), validating the potential of miniaturization of the assay. In contrast to the pronounced seeding response of the K18(P301L) interaction sensor, no change in the K18(P301L) conformation sensor signal was observed in response to most of the recombinant seeds or to brain lysates (*Figure 3—figure supplement 1D–E*). The only exception was aggK18, for which a partial decrease in the sensor signal was observed (*Figure 3—figure supplement 1D*). These results suggest that the seeds do not induce robust detectable conformational changes in the K18(P301L) sensor containing the P301L mutation and that the K18(P301L) is fully compatible with seed-induced aggregation.

Next, we raised the question of whether our WT-Tau sensor molecules, despite being resistant to the seeds from Tg brain lysates, would show increased interaction with seeding-responding forms of tau (*i.e.* K18) in the presence of pathological seeds from Tg brain lysates. For this, we assessed the complementation between LgBit-Tau and Smbit-K18(P301L) in the presence of brain lysates from WT and Tg mice. Interestingly, we observed a significant increase in the complementation signal upon treatment with brain lysates from Tg mice and not from WT mice (*Figure 3D*). This result indicates that full-length WT tau can be recruited to and interact with seeding-responsive K18 species.

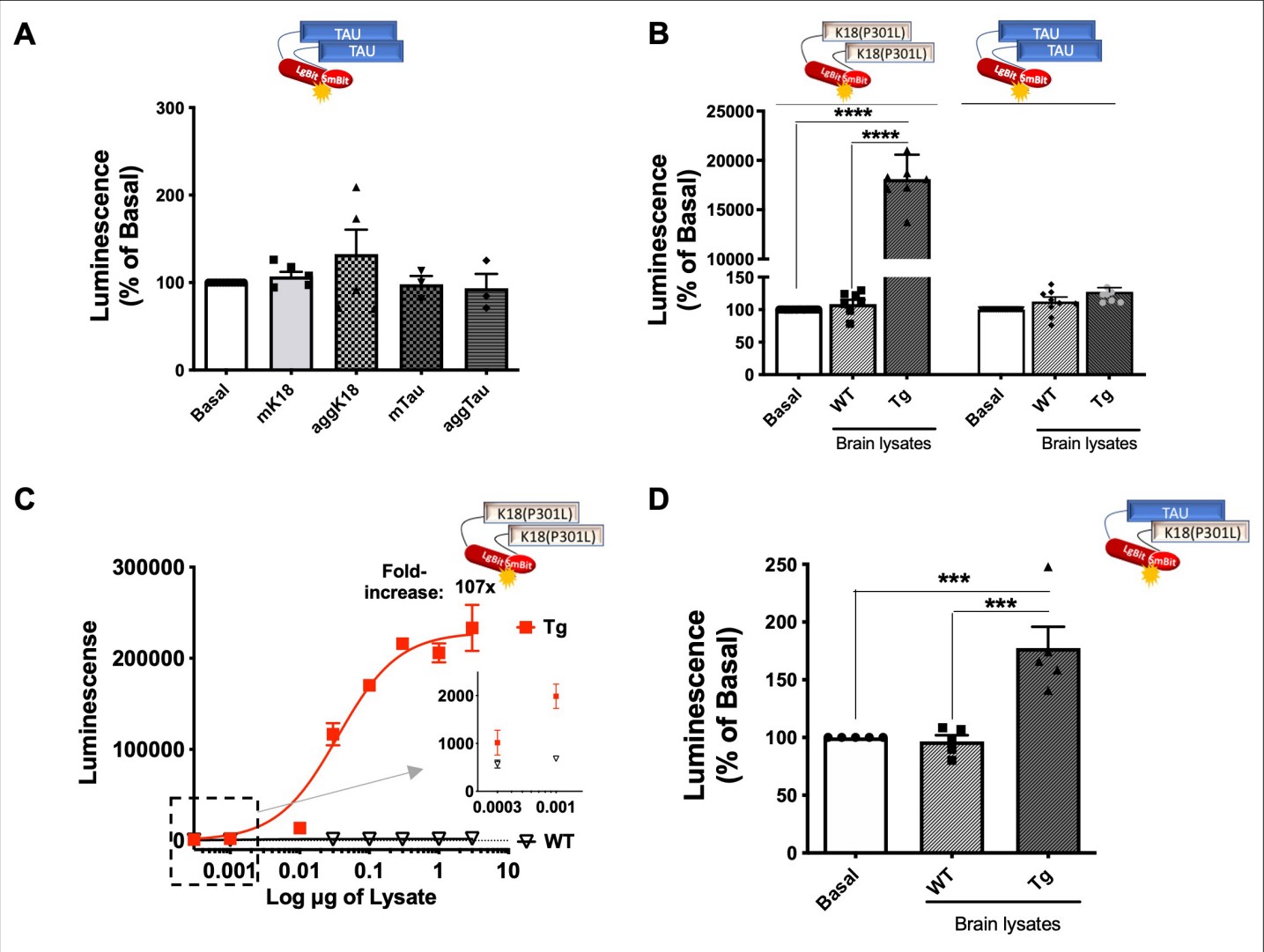

**Figure 3.** Tau seeding response monitored by Nanobinary technology (NanoBiT) tau biosensors. (**A**) WT-Tau interaction biosensor signal in cells treated with monomeric (m) or aggregated (agg) forms of recombinant K18 or tau; data are expressed as mean ± SEM of five independent experiments. (**B**) K18(P301L) and WT-Tau interaction biosensors signal in cells treated with brain lysates (3 µg of total protein, 24 hr) obtained from wild-type (WT) or transgenic (Tg) mice; data are expressed as mean ± SEM of seven independent experiments; ****p<0.001 by one-ANOVA, followed by Sidak's multiple comparison tests. (**C**) K18(P301L) interaction biosensor signal in the presence of increasing concentrations of brain lysates (in µg of total protein) obtained from WT or Tg mice; the representative curve of four independent experiments expressed as mean ± SD of triplicates. Insert: zoom of data points indicated by the square. (**D**) LgBit-Tau +SmBit-K18 interaction biosensor signal in the presence of brain lysates obtained from WT or Tg mice (3 µg); data are expressed as mean ± SEM of five independent experiments; ***p<0.005 by one-ANOVA, followed by Sidak's multiple comparison tests.

The online version of this article includes the following figure supplement(s) for figure 3:

**Figure supplement 1.** Specificity and the seeding response of the K18(P301L) conformation and interaction sensors.

### The pro-aggregating P301L mutation renders full-length tau responsive to seeding

In order to investigate whether the conformational-impacting P301L mutation would be sufficient to render full-length tau susceptible to seeding and consequently increase self-interaction, we developed full-length Tau(P301L) biosensors. The corresponding Tau(P301L) interaction biosensor was responsive to forskolin (2.5-fold increase) (*Figure 4A*), similar to the WT-Tau sensor (*Figure 2L*). The Tau(P301L) conformation biosensor showed a larger (fivefold) forskolin-induced fold-change in signal (*Figure 4B*) compared to the corresponding WT-Tau biosensor (2.5-fold) (*Figure 2K*), suggesting a higher overall phosphorylation with the concomitant higher impact of phosphorylation on the conformation of the

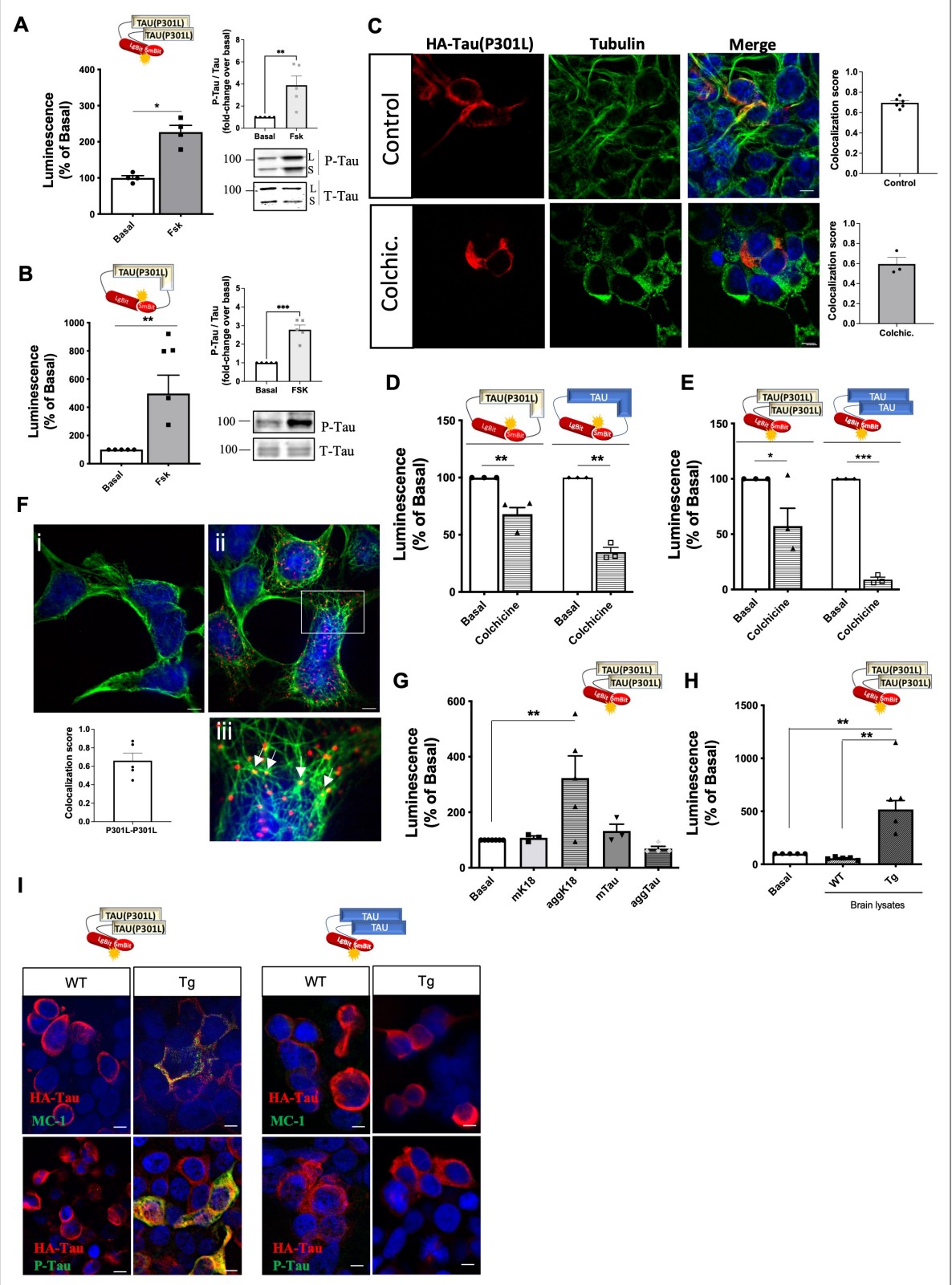

**Figure 4.** Tau(P301L) self-interaction and seeding responses monitored by Nanoluciferase (Nluc) biosensors. (**A–B**) Tau(P301L) interaction (**A**) and conformation (**B**) biosensor signal in cells treated or not with forskolin (10 μM, 24 hr); data are expressed as mean ± SEM of four (**A**) or five (**B**) independent experiments; *p<0.05, ** 0.01 by Student *t*-test. Right panel: western blot analysis of phosphorylated tau (P-Tau) and total tau (T-tau) using AT8 and anti-HA antibodies, respectively; L: Lgbit-Tau; S: SmBit-Tau. (**C**) Cellular distribution of the Tau(P301L) interaction biosensor monitored

*Figure 4 continued on next page*

*Figure 4 continued*

by immunofluorescence microscopy in the absence (top panels) or presence (bottom panels) of colchicine (biosensor: anti-HA antibody, in red; tubulin: in green; nuclei: DAPI staining, in blue); Scale bar: 10 µm. Insets: colocalization analysis (Manders' colocalization coefficient) of the fractional overlap between biosensor molecules and microtubules (MTs). (**D–E**) Tau(P301L) and WT-Tau conformation (**D**) and interaction (**E**) biosensors signal in cells treated or not with colchicine (10 µM, 1 hr); data are expressed as mean ± SEM of three independent experiments; *p<0.05, **p<0.01, ***p<0.005 by Student *t*-test. (**F**) Proximity of Tau(P301L) biosensor molecules to MTs by PLA assay (PLA signal in red; tubulin in green; nuclei in blue by DAPI staining); i. negative control (absence of primary antibodies); ii. positive PLA signal; iii. zoomed image of the white square area shown in ii; white arrows show an overlap of PLA signal and MTs; scale bar: 10 µm. Inset graph: colocalization analysis (Manders' colocalization coefficient) of the fractional overlap between PLA signal and MTs; data are expressed as mean ± SEM of five images. (**G–H**) Tau(P301L) interaction biosensor signal in cells treated for 24 hr with recombinant proteins (**G**) or brain lysates obtained from wild-type (WT) or transgenic (Tg) mice (**H**); data are expressed as mean ± SEM of four (**G**) or five (**H**) independent experiments; **p<0.01 by one-ANOVA, followed by Sidak's multiple comparison tests to the indicated group. (**I**) Immunofluorescence microscopy detection of tau pathological conformation (MC-1 antibody, green, upper panel) or phospho-tau (AT8 antibody, green, bottom panel) in cells expressing Tau(P301L) or WT-Tau interaction biosensors (HA antibody, red) and treated with brain lysates obtained from WT or Tg mice for 24 hr; nuclei are stained with DAPI (blue); scale bar: 10 µm.

The online version of this article includes the following source data and figure supplement(s) for figure 4:

**Source data 1.** Unedited blot of *Figure 4A*.

**Source data 2.** Unedited blot of *Figure 4B*.

**Figure supplement 1.** Characterization of Tau(P301L) biosensors.

Tau(P301L) sensor. Efficient and higher phosphorylation levels of the Tau(P301L) sensor (compared to WT-Tau) were confirmed by AT8 antibody immunoreactivity (*Figure 4A–B*, **inset panels**). Fluorescence microscopy analysis of the cellular distribution of the Tau(P301L) interaction sensor showed co-localization with tubulin in the cytoplasm, although to a lesser extend compared to the WT-Tau biosensor (Manders' Coefficient score 0.7; *Figure 4C*). In comparison to the K18(P301L) sensor, the staining pattern was less punctuated and absent in the nucleus. Colchicine treatment led to an intermediate phenotype, with a modest decrease in the complementation signal of both conformation and interaction Tau(P301L) biosensors and in the co-localization with MTs (*Figure 4D–E*) and in the co-localization with MTs (Manders' Coefficient score 0.6; *Figure 4C*) compared to the robust decrease of the MT-bound WT-Tau sensors and the colchicine-insensitivity of the cytosolic K18(P301L) sensors. Collectively, these observations suggest that Tau(P301L) sensors exist as a mixture of MT-bound and cytosolic tau. To further confirm this dual location of the interaction Tau(P301L) biosensor we performed the single-target proximity ligation assay (PLA). We observed PLA positive signals (labeling proximity between at least 2 sensor molecules) in cells expressing the intermolecular Tau(P301L) biosensor (*Figure 4F*, **right panel ii**), while no PLA signal is detected in the negative control (absence of the primary antibodies) (*Figure 4F*, **left panel i**). In accordance with the bioluminescence results described above, the positive PLA signals, corresponding to intermolecular tau(P301L) interaction, are detected in the cytosol (likely tau(P301L) oligomers) (*Figure 4F*; sparse red dots) as well as co-localized with MTs (*Figure 4F*, **zoom panel iii– white arrows**). With respect to the seeding response of the Tau(P301L) interaction sensor, recombinant aggK18 (*Figure 4G*) and brain lysates from Tg mice (*Figure 4H*) induced a robust response. Considering that the LgBit biosensor fusion protein has a non-negligible size, we assessed whether the LgBit tag had any interference in the properties of tau aggregation. For this, we produced the recombinant version of the LgBit-tagged Tau(P301L) protein in *E. coli* and compared its aggregation kinetics with those of the untagged Tau(P301L) produced in parallel. Aggregation kinetics measured in vitro by ThT assay showed very similar properties between the two proteins, suggesting thus that the 18kD LgBit fragment does not interfere with the intrinsic tau aggregation property (*Figure 4—figure supplement 1A*). After ruling out that the LgBit tag interferes with the aggregation properties of the tau protein, we continued the characterization of the tau intramolecular biosensor (P301L) in cells. Similar to the K18(P301L), no seeding-induced conformational change was observed for the Tau(P301L) biosensor (*Figure 4—figure supplement 1B–C*). Notably, the effect of Tg brain lysate seeds on the biosensor response correlated well with an increase in the positive immunoreactivity of the tau phosphorylation-specific AT8 antibody and the MC-1 antibody recognizing pathological tau conformations (*Jicha et al., 1997*), compared to WT brain lysate (*Figure 4I*, **left panel**). Consistently, immunoreactivity for both antibodies was undetectable in cells treated with brain lysates from WT mice and in cells expressing the WT-Tau interaction biosensor that is not responsive to either WT or Tg brain lysates (*Figure 4I*, **right panel**). In order to obtain more information on the type

of tau-tau interaction being formed with the biosensor (oligomers or fibrillar species) we performed thioflavin S (ThS) staining, which recognizes fibrillar species, together with immunostaining of the biosensor. We observed positive ThS staining only when cells were treated with Tg brain lysates, and ThS staining was not colocalized where the Tau(P301L) sensor was expressed (*Figure 4—figure supplement 1D*). A similar number of ThS-positive cells was observed in mock-transfected cells, not expressing the biosensor, indicating that the ThS staining probably originates from the fibrillar species contained in the Tg brain lysate, and that the luminescent sensor signal (measured in parallel) corresponds rather to oligomeric species that are undetectable by ThS (*Figure 4—figure supplement 1D graphs*). Taken together, these data indicate that Tau(P301L) biosensors are located at MTs and in the cytoplasm and show an intermediate phenotype between the WT-Tau and the K18(P301L) biosensors, with overall intermediate sensitivity to seeding most likely because of the pro-aggregating conformational effect of the P301L mutation.

Altogether, these results indicate that our tau biosensors efficiently report changes in tau conformational, self-interaction, or seeding responses, enabling direct comparison between the behavior of different tau forms in living cells and under different challenging contexts.

## Application of tau interaction biosensors for the identification of drug candidates

Given the established role of the seeding activity in spreading tau pathology in different brain areas, the identification of compounds able to interfere with this process holds great therapeutic potential. In view of the robust responses of our K18(P301L) and Tau(P301L) interaction biosensors towards seeds from Tg mouse brain lysates, we aimed to characterize their suitability to be used in screening for compounds interfering with their seeding/interaction response. We determined the Z-factor to estimate the high-throughput suitability and robustness of the K18(P301L) and Tau(P301L) interaction biosensor assays. Z-factor values of 0.7 and 0.5 were obtained, respectively, when comparing treatment with brain lysates from WT mice (low reference control) and Tg mice (high reference control) (*Figure 5A–B*). Z-factor values higher than 0.5 indicate that the assay displays a wide separation between the high and low references and low variability. Values between 0.7–1 are considered of excellent performance for high-throughput screenings (*Zhang et al., 1999*). As a proof-of-concept of the feasibility of using the sensor +Tg lysates to identify seeding-inhibiting compounds, we evaluated the seeding response of the K18(P301L) and Tau(P301L) interaction sensors in the presence of compounds previously reported in the literature to interfere with protein aggregation. Concomitant treatment of biosensor cells for 24 h with Tg brain lysates and compounds ID220249 (*Pickhardt et al., 2015*), CINQ-trp (*Frenkel-Pinter et al., 2017*), ID220255 (*Pickhardt et al., 2015*), and spermine (*Bera and Nandi, 2007*) had no significant effect on the Tg seed-induced complementation. In contrast, compounds ANLE138b (*Wagner et al., 2013*), leucomethylene blue mesylate (LMTMeSO4) (*Harrington et al., 2015*), and bb14 (*Messing et al., 2013*) showed a significant inhibition for both sensors (*Figure 5C–D*). Interestingly, higher inhibition efficacy was observed for the K18(P301L) sensor, suggesting that the effect of compounds was mainly dependent on the 4RD tau domain. Concentration-response curves performed with the K18(P301L) biosensor for these three compounds showed a bi-phasic inhibition curve, with a partial inhibition at nanomolar concentration and up to 80% inhibition at micromolar concentrations (*Figure 5E–G*). The pIC50 values of the first phase of the curves for each compound are as follows: ANLE138b: 10.5 ± 0.9; LMTMeSO4: 9.3 ± 0.7; bb14: 10 ± 0.9. At these $pIC_{50}$ concentrations, the three compounds inhibited the sensor signal at approximately 30%. The pIC50 of the second phase of the curves could not be determined as no plateau is reached up to the highest concentrations used. Decrease of the biosensor signal due to cell toxicity of the compounds was excluded (*Figure 5—figure supplement 1A*), as well as a non-specific effect due to interference with Nluc activity (*Figure 5—figure supplement 1B*).

Taken together, these results suggest that the NanoBiT-based tau biosensors are useful tools to monitor the behavior of the different forms of tau in a living cell context over time, and to identify or characterize candidate drugs able to interfere with the tau seeding/self-interaction processes.

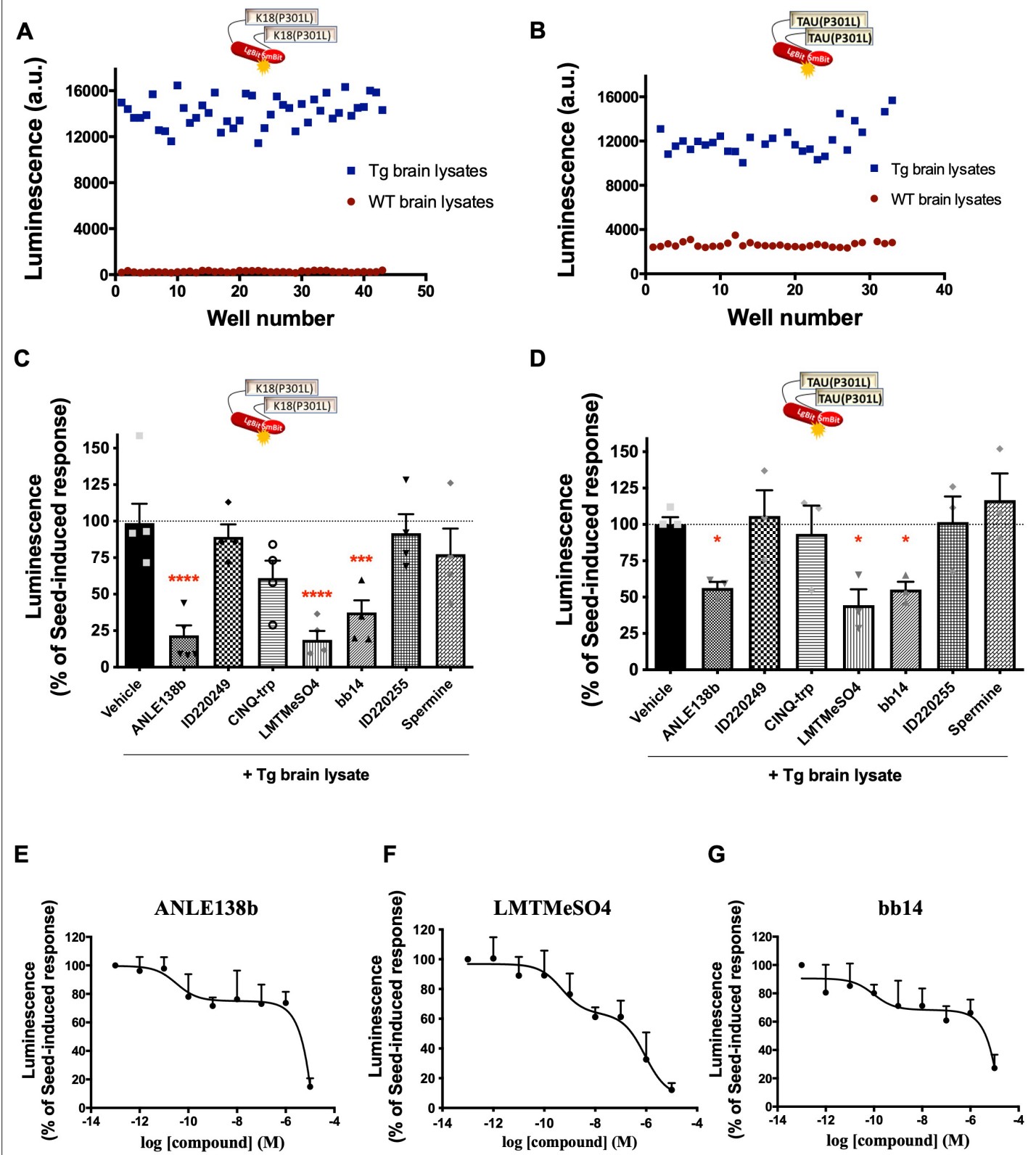

**Figure 5.** Tau Nano binary technology (NanoBiT) biosensors application in the identification of inhibitors of tau seeding and self-interaction. (**A–B**) Determination of the assay Z-factor by collecting biosensor signals for K18(P301L) (**A**) or Tau(P301L) (**B**) interaction biosensor cells in the presence of brain lysates from wild-type (WT) (red, low reference signal) or transgenic (Tg) (blue, high reference signal) mice. (**C–D**) K18(P301L) (**C**) or Tau(P301L) (**D**) interaction biosensor signal in the presence of candidate inhibitors (10 μM, simultaneously to Tg treatment, 24 hr) of tau aggregation; data are

*Figure 5 continued on next page*

*Figure 5 continued*
expressed as mean ± SEM of four (**C**) or three (**D**) independent experiments; *p<0.05, ***p<0.005, ****p<0.001 by one-ANOVA, followed by Sidak's multiple comparison tests to the vehicle group. (**E–G**) Concentration-response curves of the effect of compounds ANLE138b (**E**), LMTMeSO4 (**F**), and bb14 (**G**) on the K18(P301L) biosensor signal; data are expressed as mean ± SEM of four independent experiments.

The online version of this article includes the following figure supplement(s) for figure 5:

**Figure supplement 1.** Effect of tested compounds on Nanoluciferase (Nluc) activity and cell viability.

## Discussion

Here, we developed a series of original tau biosensors based on the NanoBiT complementation technique and demonstrate the application of these biosensors as innovative research tools able to monitor tau behavior in terms of conformational changes, self-interaction and seeding of different tau forms, in living cells and under chemical and biological challenges. These are high-throughput-compatible tau seeding/interaction and conformation assays, able to identify molecular mechanisms driving tau pathological transformation processes, as well as new drug candidates displaying potential inhibitory effects on these early tau conformation/oligomerization/spreading steps.

The molecular mechanisms initiating the transformation of tau into pathological species, especially in the case of full-length WT tau as in sporadic AD, is still poorly known. Tau aggregates in AD brains are found in a microtubule-free and hyperphosphorylated state, implying that hyperphosphorylation and microtubule dissociation are key steps preceding tau aggregation (*Braak and Braak, 1996*; *Brandt et al., 2020*). Phosphorylation is indeed a key post-translational modification of tau, regulating its dynamics of association and dissociation from MTs under physiological conditions (*Duquette et al., 2020*), and probably involved in the early conformational changes in tau pathologies (*Xia et al., 2021*; *Alonso et al., 2001*; *García-Sierra et al., 2003*). In agreement with previous reports (*Chen et al., 2019*), our tau conformational sensors evidenced that mutations like P301L impose a specific conformation that is spontaneously aggregation-prone, probably by increased exposition of the amyloidogenic domain of tau compared to WT tau. The conformational biosensors we developed here also report the conformational change of tau induced by its phosphorylation or destabilization of MTs. Our full-length WT-tau biosensors behave similar to native WT tau, such as they show similar behavior in cellular context (*i.e.* association to MTs, no detectable self-aggregation, and no detectable MC-1-reactive pathological conformation even upon exposure to potent tau seeds). In agreement with previous reports using FRET-based intramolecular biosensors (*Di Primio et al., 2017*; *Rudenko et al., 2019*), we observed that the conformation of the full-length WT-Tau is highly impacted by the destruction of MTs, most likely representing the switch from the MT-associated 'paperclip fold' structure to the unfolded structure of soluble tau (*Mandelkow et al., 2007*; *Mukrasch et al., 2009*; *Di Primio et al., 2017*). MT destruction has also a profound impact on the *inter*molecular interaction tau biosensor with a substantial loss of signal. As it is believed that tau binds to MTs in its monomeric form (*Gustke et al., 1994*), the basal WT-Tau biosensor signal most likely reports on the proximity of tau molecules at MTs, which is lost upon MT destruction, rather than on direct tau-tau self-interaction. This conclusion is highly relevant in view of recent attempts to identify inhibitors of WT-tau interaction based on the inhibition of the basal signal observed in cell-based protein interaction assays (*Lo et al., 2019*; *Holzer et al., 2018*). Based on our findings, such inhibitors possibly detach tau from MTs and would be therefore rather detrimental to tau physiological functions. Accordingly, our WT-tau sensor can be considered as a control assay, to ensure that compounds of interest do not destabilize the tau-MT interaction. In addition, our full-length WT-tau sensors provided further important insights on tau behavior by suggesting that phosphorylation of WT tau takes place when it is still bound to MTs, as colchicine and OA prevented its phosphorylation-induced sensor response. Both intra- and intermolecular tau biosensors are, thus, useful tools to elucidate the mechanisms underlying the pathological transformation of native non-mutated tau by identifying cellular stressors and/or signaling pathways linked to age-related tau conformation change and/or dissociation from MTs. Interestingly, our K18(P301L) interaction sensor showed a punctuated expression pattern and cytoplasmic and nuclear localization. Previous reports on a fluorescence-based aggregation sensor comprising K18-YFP also showed nuclear localization of the sensor (*Sanders et al., 2014*). Similar nuclear localization was observed for P301S 0N4R tau-YFP sensor (*Lester et al., 2021*), while our full-length P301L 2N4R tau sensor did not reveal prominent nuclear localization neither before nor after seeding. Whether the

cytoplasmic:nuclear cellular distribution of tau depends on its N-terminal domain (2 N vs. 0 N) is an interesting hypothesis that merits future investigation in view of the recent evidence of the role of tau on nuclear RNA processing (*Lester et al., 2021*).

Propagation of tau pathology from cell-to-cell through a prion-like mechanism has been now extensively described and it relies on the ability of pathological tau forms to transmit its conformation and self-interaction behavior to other tau molecules from neighboring cells, spreading the pathology to several brain areas (*Frost et al., 2009*). Our K18(P301L) and Tau(P301L) interaction biosensors are highly responsive to various seeds. A side-by-side comparison between ThS staining, revealing the formation of fibrillar structures, and the biosensor signals indicate that the sensor signal does not correlate with ThT staining, strongly suggesting that the biosensor signal originates rather from oligomeric than fibrillar forms of tau. In vitro aggregation studies with purified LgBit-Tau(P301L) indicate that the sensor is actually able to form fibrils to a similar extend as the untagged Tau(P301L), excluding thus the possibility that the sensor is intrinsically unable to form fibrils. In contrast, and in accordance with the literature, our WT-tau biosensor turned out to be highly resistant to oligomerization, even in the presence of potent tau seeds present in the brain lysates of P301S mouse model. Interestingly, the presence of SmBit-K18(P301L) engaged WT LgBit-Tau in a common complex which was further increased in the presence of seeds. This result suggests that WT tau molecules do not self-interact easily, but they can interact with and be sequestered into oligomers formed by seeding-responsive tau species. This new biosensor comprising K18 and WT-tau full-length might represent an interesting new target for the identification of therapeutic molecules preventing the seeding of physiological tau molecules.

Previously developed cell-based tau oligomerization biosensors were mainly based on tau conjugated to fluorescent proteins or on bi-molecular fluorescence complementation (BiFC) technique for visualization of tau aggregates by fluorescent microscopy (*Chun et al., 2007*; *Tak et al., 2013*), or FRET (*Holmes et al., 2014*; *Lo et al., 2019*; *Di Primio et al., 2017*). The first assays on the luciferase complementation technology applied to tau aggregation were based on *Renilla* or *Gaussia luciferase* complementation assays (*Mirbaha et al., 2015*; *Sanders et al., 2014*; *Wegmann et al., 2016*; *Wang et al., 2017*). The quantitative properties of these assays were essential to determine the minimum functional unit of tau seeds required to induce self-interaction of a K18-based complementation sensor (*Mirbaha et al., 2015*), as well as to characterize the seeding activity of distinct tau strains (*Sanders et al., 2014*) and tau-containing exosomes (*Wang et al., 2017*). To the best of our knowledge, our K18(P301L) and tau(P301L) full-length sensors are the first NanoBiT-based tau seeding assays. As previously mentioned, among the main advantages of NanoBit-based sensors are the smaller size of the sensor part (compared to previous FRET or fluorescent sensors), their quantitative properties, easy set up in a plate reader, and straightforward data analysis. In accordance with previous reports (*Mirbaha et al., 2015*; *Sanders et al., 2014*; *Wegmann et al., 2016*; *Wang et al., 2017*), we also observed that sensors comprising only the K18 fragment show the best signal amplitude in response to extracellular seeds, including those obtained from brain lysates of a Tg mouse model of tauopathy. The study from Wegmann and collaborators (*Wegmann et al., 2016*) was the only one to use full-length WT tau and, similarly to our results, no significant self-interaction of the sensor was observed in the presence of brain homogenates from the transgenic P301S mouse model, confirming the resistance of WT-Tau to be seeded compared to K18 or mutated full-length tau. The sensitivity of our K18(P301L) interaction sensor is remarkable, spanning three logs of protein concentration of brain lysates, being able to discriminate between brain lysates from control and transgenic mice in as low as 300 pg of total protein after only 24 hr of incubation, and responding to subnanomolar concentrations of recombinant K18 seeds. The Nluc complementation-based Tau(P301L) and K18(P301L) interaction assays presented here are of good sensitivity for the detection of pathological seeds with a 3-orders- of-magnitude, similar assay window described for a previous FRET-based K18 aggregation sensors (*Holmes et al., 2014*; *Lo, 2021*; *Hitt et al., 2021*), (taking into account lower incubation time in our study) and are, in addition, high-throughput compatible and rely on a simple and fast luminescence readout.

Lastly, we provide 'proof-of-principle' evidence of the robustness of both Tau(P301L) and K18(P301L) interaction assays and their application in the identification of small molecules able to interfere with tau seeding/self-interaction response, opening novel possibilities for the discovery of new drug candidates for tau-related neurodegenerative diseases. Among the seven compounds

tested for this proof-of-concept study, three (ANLE138b, bb14, LMTMeSO4) were effective in inhibiting the tau seeding response of our interaction sensors. All three showed a biphasic concentration-dependent effect. The reason for this biphasic behavior is currently unknown but might represent the interference with two different oligomeric tau forms, which still needs to be further characterized. ANLE138b showed an 85% decrease in our biosensor assay at 10 μM. ANLE138b was previously described to inhibit α-synuclein and tau aggregation in a cell-free in vitro aggregation assay with an $EC_{50}$ of 2.8 μM for α-synuclein and a partial inhibition (approximately 30%) at 10 μM for tau (; *Wagner et al., 2015*). In vivo, ANLE138b ameliorated neuropathology and cognitive behavior and reduced the amount of large tau aggregates in the brain of PS19 tauopathy mouse model (*Wagner et al., 2015*). The rhodamine-compound bb14 inhibited tau aggregation by 70% in our assay at 10 μM. Bb14 was previously reported not only to prevent tau aggregation ($IC_{50}$ = 0.67 μM in ThS assay), but also to disaggregate pre-formed tau fibrils ($IC_{50}$ = 0.94 μM) (*Bulic et al., 2007*). In hippocampal organotypic slices from transgenic pro-aggregant $Tau_{RD}\Delta K280$ mice, bb14 decreased ThS-positive or sarkosyl-insoluble tau fibrils by 70% at 15 μM after treatment for several days (*Messing et al., 2013*; *Bulic et al., 2007*). LMTMeSO4, a second generation of methylene blue derivative, displayed the most interesting profile in our seeding assay, with a 40% inhibition at 10 nM and 90% inhibition at 10 μM. LMTMeSO4 was previously shown to inhibit fibril formation in various cellular and in vitro aggregation assays with $EC_{50}$ values ranging from 0.16 to 238 μM (*Harrington et al., 2015*). Our own data complement these observations and indicate an effect of LMTMeSO4 on the initial oligomerization/seeding process of tau. LMTMeSO4 (TRx0237 from TauRx therapeutics) is currently under phase III clinical trial in a low-dose protocol targeting early AD patients (NCT03446001). Taken together, our results suggest that ANLE138b, bb14, and LMTMeSO4 may interfere with the seeding response detected by our biosensors in addition to the previously described interference with tau fibril formation. Additional effects of these compounds on preformed seeds present in brain lysates cannot be excluded at this stage but have to be considered in further studies aiming at the full characterization of their properties. In summary, we developed a series of highly sensitive NanoBiT tau biosensors able to monitor tau conformational changes and self-interaction/seeding responses in living cells exposed to different challenges related to diseases development, including over-activity of kinases, MT disruption, or the presence of pathological tau seeds. Our study is the first one that generated and systematically compared the response of different tau forms in terms of conformational changes and self-interaction towards different challenges in a cellular context. These sensors comprise a platform for the investigation of tau physiological and pathological behaviors, as they are useful tools to better understand the diversity of response of different tau forms, the cellular conditions that might favor tau pathological transformation and seeding, as well as to be applied in high-throughput screening programs to the identification of potential drug candidates disrupting tau seeding and self-interaction responses.

## Methods

**Key resources table**

| Reagent type (species) or resource | Designation | Source or reference | Identifiers | Additional information |
|---|---|---|---|---|
| gene (*Homo sapiens*) | MAPT | GenBank | NCBI Entrez Gene: 4137 | |
| strain, strain background (*Escherichia coli*) | BL21(DE3) | Sigma-Aldrich | CMC0016 | Electrocompetent cells |
| cell line (*Homo sapiens*) | Human Embryonic Kidney (HEK) 293 | Sigma-Aldrich | RRID: CVCL_0063 | |
| cell line (*Homo sapiens*) | Human neuroblastoma SH-SY5Y cells | Sigma-Aldrich | RRID:CVCL_0019 | |
| transfected construct (Tag constrcut) | NanoBiT system (SmBit and LgBit) | Promega | | |
| transfected constructo (*Homo sapiens*) | LgBit-HA-K18(P301L) | This paper | | see Methods |
| transfected construct (*Homo sapiens*) | SmBit-HA-K18(P301L) | This paper | | see Methods |

*Continued on next page*

*Continued*

| Reagent type (species) or resource | Designation | Source or reference | Identifiers | Additional information |
|---|---|---|---|---|
| transfected construct (*Homo sapiens*) | LgBit-HA-Tau(P301L) | This paper | | see Methods |
| transfected constructo (*Homo sapiens*) | SmBit-HA-Tau(P301L) | This paper | | see Methods |
| transfected constructo (*Homo sapiens*) | LgBit-HA-Tau | This paper | | see Methods |
| transfected constructo (*Homo sapiens*) | SmBit-HA-Tau | This paper | | see Methods |
| antibody | anti-HA antibody (rabbit monoclonal) | Cell Signaling | Cat# 3724 S | IF(1:500), WB (1:1000) |
| antibody | anti-HA antibody (mouse monoclonal) | Biolegend | Cat#. 901514 | IF(1:500) |
| antibody | anti-tubulin (rat polyclonal) | Millipore | Cat#: MAB1864 | IF(1:200) |
| antibody | anti-phospho tau antibody AT8 (Mouse monoclonal) | Thermofisher | MN1020; | IF(1:200), WB (1:500) |
| antibody | anti-aggregated tau MC-1 antibody (Mouse monoclonal), conformation-specific anti-tau antibody | PMID:9349554 | | IF(1:500) |
| antibody | IRDye 800CW anti-Rabbit IgG (Goat polyclonal) Secondary Antibody | LI-COR Biosciences - GmbH | Cat.# 926–32211 | WB (1:10,000) |
| antibody | IRDye 680RD anti-Mouse IgG (Goat polyclonal) Secondary Antibody | LI-COR Biosciences - GmbH | Cat.# 926–68070 | WB (1:10,000) |
| commercial assay or kit | nanoluciferase substrate Nano-Glo Live Cell | Promega | Cat.# N2012 | |
| commercial assay or kit | Duolink PLA | Sigma-Aldrich | Cat.# DUO92101 | |
| chemical compound, drug | thioflavin S | Sigma Aldrich | Cat.# T1892 | |
| chemical compound, drug | thioflavin T | Sigma Aldrich | Cat.# T3516 | |
| software, algorithm | Image J | https://doi.org/10.1038/nmeth.2089 | RRID: SCR_003070 https://imagej.nih.gov/ij/download.html | |
| software, algorithm | GraphPad Prism 6 | GraphPad Software Inc | RRID:SCR_002798; https://www.graphpad.com/scientificsoftware/-prism/ | |
| other | EnVision Plate Reader | PerkinElmer | | see Methods |

## Expression vectors

Plasmids encoding for NanoBiT system (SmBit and LgBit) were purchased from Promega (Madison, USA) and cloned upstream of the tau protein in pcDNA3.1 vectors by restriction enzymes. The tau constructs used in this study comprise: (**i**). tau full-length (aa 1–441 of 2N4R tau); (**ii**). tau(P301L) (tau full-length displaying the P301L mutation); (**iii**). K18(P301L) (aa 244–369 comprising the four repeat domains of tau with the P301L mutation). The tau P301L mutants were generated by PCR using QuickChange Lightning Site-Direct mutagenesis kit (Agilent Technologies, Santa Clara, USA). For all intermolecular sensors, we generated N- and C-terminal LgBit/SmBit conjugated tau constructs. All the combination pairs were tested and the best configuration for each biosensor was chosen based on the criteria of displaying a measurable basal signal. Similarly, LgBit and SmBit at either N- or C-terminal positions were also tested for all intramolecular sensors. All constructs include an HA-tag and their sequences were verified by DNA sequencing.

## Cell culture and cell transfection

HEK293T (Sigma-Aldrich; RRID: CVCL 0063; authenticated by the provider) were maintained in Dulbecco's Modified Eagle's Medium (DMEM) Glutamax (Invitrogen) supplemented with 4.5 g/L glucose, 10% fetal bovine serum, 100 U/mL penicillin, and 0.1 mg/mL streptomycin, at 37 °C (95% $O_2$, 5% $CO_2$). SH-SY5Y cells were maintained in DMEM:F12 Medium (Invitrogen) supplemented with 10% fetal bovine serum, 100 U/mL penicillin, and 0.1 mg/mL streptomycin, at 37 °C (95% $O_2$, 5% $CO_2$). Cell lines were checked regularly for mycoplasma contamination. Cells were transfected with equal amounts of LgBit and SmBit construct (250 ng each per well in 6-well plates; or 125 ng each per well in 12-well plates) or 250 ng (6-well plates) of the intramolecular sensor. For the saturation curves, specific amounts of each construct were used, as indicated. Transfections were performed using jetPEI reagent according to the supplier's instructions (Polyplus-transfection, New York, NY, USA). One day after transfection, cells were plated into white 96-well plates (Perkin Elmer Life Sciences, Waltham, MA) precoated with 10 µg/ml poly-L-lysine (Sigma-Aldrich), and the Nluc complementation assay was performed 48 hr post-transfection.

## Preparation of recombinant proteins and mouse brain lysates

His6-K18 (244-372), His6-WT 2N4R tau, tau(P301L), and LgBit-Tau(P301L) were produced in *E. coli* BL21 (DE3) bacteria and purified using IMAC capture followed by ion exchange chromatography (GTP biotechnology). K18 protein was eluted in 20 mM sodium phosphate, 500 mM NaCl, pH 7, 1 mM EDTA, and 2 mM DTT. The 2N4R tau protein was eluted in 25 mM sodium phosphate, 25 mM NaCl, 2.5 mM EDTA pH 6,6, and lyophilized in 50 mM 'ammonium acetate pH 8.2. Tau(P301L) and LgBit-Tau(P301L) proteins were eluted in phosphate-buffered saline and 1 mM DTT. Aggregated species of recombinant tau full-length and K18 fragments were prepared as previous reported (*Mirbaha et al., 2015*). Briefly, monomeric species were preincubated in 10 mM dithiothreitol for 60 min at room temperature, followed by incubation in oligomerization buffer (10 mM HEPES, 100 mM NaCl, and 2–6.8 µM heparin (1:1 ratio of monomeric tau to heparin, 37 °C, 24 hr)). Oligomers of amyloid beta peptide were obtained from lyophilized $A\beta_{1-42}$ (Anaspec), which was firstly dissolved in 1,1,1,3,3,3-hexafluoro-2-propanol, evaporated, aliquoted and reconstituted in anhydrous DMSO and PBS, followed by incubation for 24 hr at 4 °C prior to use (*Cecon et al., 2019*). Aggregated state of the preparations was assessed by thioflavin T assay. Brain-derived tau seeds were obtained from the P301S transgenic mice model of tauopathy, as their brain lysate has been shown to facilitate the formation of tau-tau interaction in HEK cells (*Holmes et al., 2014*). Heterozygous mice expressing human tau 1N4R with the P301S mutation (*Yoshiyama et al., 2007*), and WT littermates (referred to as Tg and WT mice, respectively) were obtained from Jackson Laboratories (Stock No: 008169) and bread for SERVIER at Charles River (Lyon, France). Five Nine-month-old male Tg and WT mice were sacrificed, and the cortex was rapidly removed and flash-frozen. Frozen tissue was lysed in 100 mg/mL of PBS (-$Ca^{2+}$, -$Mg^{2+}$), using a cooled Precellys homogenizer. Lysates were sonicated (Q800R2, Sonica) at 4 °C for 1 min at 90% power, and centrifuged for 15 min at 21.000 g, 4 °C. The supernatant was collected, aliquoted, and stored at –80 °C. Protein concentration was determined using a Pierce BCA Protein Assay Kit (ThermoFisher). Prior to use, lysates were sonicated again in a probe sonicator (Branson SLPe, 90%, 6 × 10 s, with 10 s intervals). Sample sizes were designed to give statistical power while minimizing animal use.

## Nluc complementation assay

Transfected cells expressing tau sensors were incubated with forskolin (10 µM, 24 hr) and/or colchicine (10 µM, 1 hr), in the 96-well plate. For tau seeding experiments, recombinant tau proteins (100 nM, 24 hr) or brain lysates from Tg and WT mice (up to 3 µg of total protein/well, 24 hr) were mixed with Lipofectamine 2000 reagent (1 µL/well) in Opti-MEM (Invitrogen) and incubated for 20 min before adding to HEK293T cells, as previously reported (*Holmes et al., 2014*). Immediately before the assay, the medium was removed, washed once with PBS, and replaced with PBS. The nanoluciferase substrate Nano-Glo Live Cell (Promega) was added to the wells at a final dilution of 1:100 and luminescence signal was read immediately and for several cycles in order to assure the detection of the plateau of the enzyme:substrate reaction, using the EnVision plate reader (PerkinElmer). The data acquired at the plateau phase (at 10 min after the addition of the substrate) were used for all graphs unless otherwise stated.

## Immunofluorescence analysis

HEK293T cells transiently expressing each sensor were seeded onto sterile poly-L-lysine coated 24-well format cover glass one day after transfection with the indicated tau biosensors, and treated with the indicated compounds. At the end of the incubation time, cells were washed, fixed with paraformaldehyde 4% or methanol:acetone (1:1) solution (15 min), permeabilized in 0.2% Triton X-100, and blocked for 1 hr with 5% horse serum. Cells were then incubated (overnight, 4 °C) with monoclonal anti-HA antibody (rabbit, dilution 1:500, cat. 3724 S; Cell Signaling), or anti-tubulin (rat, MAB1864; Millipore; 1:200), or with anti-phospho tau antibody AT8 (MN1020; Thermofisher; 1:200), or with the tau conformation-specific MC-1 antibody (dilution 1:500, *Jicha et al., 1997*). MC-1 antibody was generously provided by Dr. Peter Davies. After several washes, cells were incubated with anti-mouse or anti-rat, or anti-rabbit fluorescein-conjugated secondary antibodies (dilution 1:500, Invitrogen) and cell nuclei were stained with DAPI (SigmaAldrich). After mounting the cover glass into glass slides, the slides were analyzed under Zeiss Observer Z1 microscope (Zeiss, Germany). Images obtained were further analyzed using ImageJ software (*Schneider et al., 2012*). Colocalization was assessed by measuring the fractional overlap in at least three different images, in several fields comprising at least two cells positive for sensor expression, by Manders' Colocalization Coefficient analysis using the JACoP plugin for ImageJ.

## Proximity ligation assay (PLA)

Biosensor-expressing cells plated in 8-well chamber slides (ibidi) were fixed in methanol:acetone (1:1) and permeabilized as described for immunofluorescence analysis. All the following steps of the proximity ligation assay were performed according to suppliers' instructions (Duolink PLA; Sigma-Aldrich). Tau-tau sensor molecules interaction was assessed using two monoclonal antibodies against the HA tag (mouse, Biolegend cat. 901514; rabbit, Cell Signaling 3724 S), and two PLA probes (anti-rabbit-PLUS, anti-mouse-MINUS). Tubulin and nucleus staining was performed after the PLA as described in the immunofluorescence section and the slides were analyzed under Zeiss Observer Z1 microscope (Zeiss, Germany). Images obtained were further analyzed using ImageJ software.

## Thioflavin S staining

Immediately after the immunostaining procedure with the anti-HA antibody, cells were incubated with 0.1% ThS (Sigma-Aldrich) in 50% ethanol solution for 15 min, followed by two washes of 20 min with 50% ethanol and one wash with 80% ethanol. After one final wash in PBS, the cover glasses were mounted into glass slides and the slides were analyzed as described in the immunofluorescence analysis section (ThS ex: 391/em: 428 nm).

## Thioflavin T assay

In vitro aggregation assay was performed by incubating tau or Aß recombinant proteins (10 µM) in buffer containing 10 mM HEPES, 100 mM NaCl, 2 mM DTT, and 50 µM thioflavin T (Sigma-Aldrich), in the presence or absence of oligomerization inducer (heparin:tau at 1:1 concentration) in a black 96-well plate. Thioflavin T fluorescence was read immediately and every 30 min for 24 hr or up to 120 hr using the Envision plate reader (ex: 430 nm; em: 480 nm; with a 30 s shaking step before each read).

## SDS-PAGE/Western blot

Cells expressing tau biosensors were lysed in Laemli buffer (62.5 mmol/L Tris/HCl pH 6.8, 5% SDS, 10% glycerol, 0.005% bromophenol blue) and resolved in SDS-PAGE gel (10 or 12%), followed by protein transfer to nitrocellulose membranes. Membranes were blocked in 5% non-fat dried milk in TBS (10 mM Tris-HCl, pH 8, 150 mM NaCl), and immunoblotted with primary antibodies against the HA tag (rabbit, dilution 1:1000, cat. 3724 S; Cell Signaling), or anti-phospho tau antibody AT8 (MN1020; Thermofisher; 1:200), diluted in 0.3% BSA in TBS (overnight, 4 °C), followed by incubation with secondary antibodies coupled to 680 or 800 nm fluorophores (1:15,000 dilution in 0.3% nonfat dried milk in TBST (TBS with 1% Tween 20), 1 hr at room temperature; LI-COR Biosciences, Lincoln, NE, USA). Membranes were read using the Odyssey LI-COR infrared fluorescent scanner (LI-COR Biosciences). Blot quantifications were performed using Empiria Studio software (LI-COR Biosciences).

## Data analysis

Data are presented as means ± SEM of the indicated $n$ (number of independent experiments). A minimum of three independent experiments, each performed in triplicates (technical replicates), were performed to an ensure accurate estimation of experiment variability. Nluc complementation experiments were each performed in triplicates to ensure the reliability of single values, and luminescence data are expressed as the raw data or normalized to % of basal signal or maximal signal, as indicated. Non-specific complementation signal was determined for inter-molecular sensors by expressing only one of the constructs or co-transfecting with a complementation construct comprising a protein (halo) non-related to tau. For the experiments on concentration-response curves of inhibitory compounds, data were fitted to the nonlinear regression log (inhibitor) versus response (three parameters) fitting equation (GraphPad Prism software version 6). Statistical analysis was performed using GraphPad Prism software version 6, applying Student's $t$-est to compare two groups or ordinary one-way or two-way analysis of variance (ANOVA) when comparing multiple groups, followed by Tukey's or Dunnett's multiple comparison post hoc test when appropriate. Values of p<0.05 were considered statistically significant. Z-factor values (*Zhang et al., 1999*) of the Nluc complementation assay for the K18(P301L) and Tau(P301L) interaction biosensors were determined using the following equation, with cells treated with brain lysates from WT mice used as negative control and cells treated with Tg brain lysates as positive control:

$$\text{Z-factor} = 1 - \frac{3\left(\sigma_p + \sigma_n\right)}{\left|\mu_p - \mu_n\right|}$$

where σ$P$=standard deviation of positive control; σn=standard deviation of negative control; μ$P$=mean of positive control; μn=mean of negative control.

## Acknowledgements

The authors are grateful to members of the Jockers laboratory for stimulating discussions and comments on the project. We thank Aurelien Cordier for his expert technical help in the initial phase of the project. We thank the support from the IMAG'IC core facility of Cochin Institute (Paris, France). The work was supported by Servier Laboratories, Institut National de la Santé et de la Recherche Médicale (INSERM), and Centre National de la Recherche Scientifique (CNRS). We thank the support from the Fondation Philippe Chatrier.

## Additional information

### Funding

| Funder | Grant reference number | Author |
| --- | --- | --- |
| Institut National de la Santé et de la Recherche Médicale | | Ralf Jockers |
| Centre National de la Recherche Scientifique | | Ralf Jockers |
| Servier | | Patricia Machado |

The funders had no role in study design, data collection and interpretation, or the decision to submit the work for publication.

### Author contributions

Erika Cecon, Conceptualization, Formal analysis, Investigation, Writing – original draft, Writing – review and editing; Atsuro Oishi, Conceptualization, Investigation; Marine Luka, Delphine Ndiaye-Lobry, Arnaud François, Mathias Lescuyer, Investigation; Fany Panayi, Conceptualization; Julie Dam, Conceptualization, Writing – original draft, Writing – review and editing; Patricia Machado, Conceptualization, Writing – review and editing; Ralf Jockers, Conceptualization, Supervision, Funding acquisition, Writing – original draft, Writing – review and editing

## Author ORCIDs
Erika Cecon ⬥ http://orcid.org/0000-0003-2387-9313
Atsuro Oishi ⬥ http://orcid.org/0000-0003-1788-5485
Ralf Jockers ⬥ http://orcid.org/0000-0002-4354-1750

## Ethics
Ethics Committee ApprovalAll animal experimental procedures were performed in agreement with the European Communities Council Directive 2010/63/EU and were approved by the Ethical Committee for Animal Experimentation of the Institut de Recherches Servier.

## Decision letter and Author response
Decision letter https://doi.org/10.7554/eLife.78360.sa1
Author response https://doi.org/10.7554/eLife.78360.sa2

---

# Additional files

## Supplementary files
• Transparent reporting form

## Data availability
All data generated or analysed during this study are included in the manuscript and supporting file; Source Data files have been provided for blots in Figures 1, 2 and 4.

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
