## [Editor Report]

Understanding specific tau-tau interactions that play key roles in Alzheimer's disease and tauopathies will enable the elucidation of the toxic tau species involved in the pathogenesis of these diseases and therapeutic development in this area. In their important paper, Cecon et al. develop a series of NanoBiT complementation-based tau biosensors to monitor tau intramolecular and intermolecular interactions. This solid work will be of high interest to a broad target audience including researchers in the field of biophysics, biochemistry, cell biology, neuroscience, neuropathology, and drug discovery.

---

## [Decision Letter]

**Decision letter after peer review:**

Thank you for submitting your article "Novel repertoire of tau biosensors to monitor pathological tau transformation and seeding activity in living cells" for consideration by *eLife*. Your article has been reviewed by 3 peer reviewers, and the evaluation has been overseen by a Reviewing Editor and Anna Akhmanova as the Senior Editor. The following individual involved in review of your submission has agreed to reveal their identity: Sylvie Claeysen (Reviewer #1).

Essential revisions:

Overall, while the reviewers see potential in this paper, substantial revisions are required. I urge you to go through the reviewers' comments thoroughly, as they bring up similar concerns about the need for better characterization of the biosensors, important controls, and toning down language to make the conclusions more reflective of the results.

1) All three reviewers brought up concerns about the strong claims stated within the paper on how this assay is better than other currently available assays without providing any data to this effect. The authors must discuss other biosensors more thoroughly and either tone the language down substantially or repeat some of their experiments with other split-luciferase or fluorescent biosensors.

2) Can the authors use other measurements or assays to show that tau aggregation is occuring on the microtubule, and perhaps disrupt the microtubule network using another method such as nocodazole to show a decrease in tau aggregation? Please also include the immunofluorescence images of tubulin and tau both individually and merged with colchicine treatment. It would also be a good control to show that colchicine is not directly disrupting tau-tau interactions or affecting the phosphorylation state of tau.

3) All of the reviewers agreed that there must be better characterization of the biosensors. The reviewers suggested that the authors should demonstrate that interaction and aggregation kinetics are not affected significantly compared to the native tau protein in vitro to support the physiological relevance of the claims related to inter- and intra-molecular interactions. Are there ways to differentiate between intramolecular and intermolecular interactions other than the luciferase signals? The colocalization data is not of high enough quality to support the claims regarding microtubule interactions, and a reviewer pointed out that there seems to be stronger colocalization with the intramolecular sensor than with the intermolecular one. Better quality images and co-localization analysis are needed to support these interpretations. A more extensive evaluation of the sensing capacity would be needed to establish sensitivity in a meaningful way, for instance with tau proteins for which concentration can be more appropriately estimated, e.g., recombinant tau and IP-purified extracts from mouse and human tissues, or a direct comparison with other methods. Finally, is it possible to further assay what kind of tau-tau interaction are being formed and recognized by the biosensors: are they oligomers or fibrillar species? Perhaps ThS staining could provide more information on the type of interactions.

4) The reviewers agreed that the data related to the effect on phosphorylation needs further validation. It was noted that the effects of forskolin are subtle based on the blots shown, and perhaps other treatments that affect kinase activity could be used to confirm the findings. This should also be taken into account when employing the combination treatment with colchicine. The authors should provide blots of phosphorylation status upon colchicine treatment.

5) The reviewers were concerned about how the authors can use the biosensors directly to screen for small molecules that affect tau aggregation with their current biosensor as a read-out. The authors should perform Z-factor calculations on the small molecules as positive controls with direct relevance to the biosensors and not indirectly through affecting the Tg brain lysates. The authors should also calculate the IC50 of the compounds tested and compare their values with those in the literature, as well as clearly state the concentrations they are using. There was also a recommendation that the authors test methylene blue, which is in phase 3 for its anti-tau aggregation properties.

*Reviewer #1 (Recommendations for the authors):*

Some recommendations and suggestions are provided below to improve the quality of this very interesting work.

1) The suitability of tau biosensors for high-throughput screening needs to be further developed as experiments were only performed in 96-well plates. No scaling up in 384- or 1536-well plates was attempted. Moreover, the transfection of the biosensors could be tedious and the generation of a stable cell line could have been envisaged.

2) The models used are basic (HEK-293T cells associated with tau aggregates, Aβ oligomers or mouse brain lysates). We might regret that no other cellular model of neurodegenerative disease has been considered, such as the lentiviral expression of biosensors in an IPS cell line derived from a patient carrying a tau mutation. The number of drugs tested is also quite low. Methylene blue, which is tested in phase 3 for its anti-tau aggregation properties, could have been tested.

Results

Page 6, line 23: The nuclear localization of the biosensor is quite substantial and must be commented on.

Page 7, lines 1-7: A Western Blot presenting the aggregated products (K18, Full-length tau, Aßo) used in comparison with the non-aggregated forms could be presented. In addition, please provide indications regarding the quantity of proteins and Aßo that were used in the experiments.

Figures 2C, 2D and 4D, 4E and corresponding text: Visualizing by microscopy the effect of colchicine treatments could reinforce the demonstration.

Figure 2E: Please indicate error bars on both sides to be consistent with Figure 2A.

Page 9, line 2: Please correct P301L.

Page 9, line 13: Please briefly explain the mode of action of forskolin (PKA activator) and edit line 16 as it sounds like forskolin itself is responsible for tau phosphorylation.

Figures 2I, 2J and 4A, 4B, and corresponding text: Please provide quantification of the blots showing phosphorylation of tau biosensors.

Page 9, lines 22-23: Co-treatment with colchicine followed by forskolin was performed on interaction WT-Tau sensors. What is the effect on the conformation WT-Tau sensors?

Page 9, line 25: "to monitor the behavior of tau in a physiological cellular context" and Page 17 line 1 "physiological context": In my opinion, this is not a real "physiological" cellular context as the biosensors have been overexpressed in the cells. I would rather use the term "living cell" This also raise the point of the amount of tau biosensor versus the level of endogenous tau. Did the authors verify this point using tau antibodies recognizing both endogenous tau and biosensors?

Figure 3 supplement 1 and 2: I would merge these two figures, and use more conventionally the terminology of Supplementary Figures 1 to 4, unless it is specific the Journal.

Figure 4H left panel and corresponding text: How does the authors explain that not all cells expressing the biosensors show MC1 staining? Could this be based on an induction threshold for MC1 staining? Regarding AT8 staining, it would be preferable to present an image showing several cells.

Methods

Page 20, line 5: Contrary to what is indicated here, the amount of each construct is not always indicated. For reproducibility purpose, the amount of cDNA transfected should be indicated in the legend of each figure panel, along with the number of cells used.

Page 20, line 17: Please replace "Phosphate de sodium" by "Sodium phosphate".

Page 20, line 18: Please replace "acétate d'ammonium" by "ammonium acetate".

Page 21, line 6: Please indicate how many mice were used. In the case of pooled extracts, how many different mice were mixed in the pools?

Page 21, line 20: The "micro" (µ) signs have been replaced by u signs.

Page 22, line 2: Please mention here that an acquisition time of 10 minutes was used for all graphs, unless otherwise stated.

Page 23, line 5: Please define TBST.

Page 24, line 5: The authors indicate that they have calculated the Z-prime (Z', calculated from the medians) but they indicate the formula of the Z-factor (calculated from the means). Please clarify.

Figures

According to *eLife* policies, "Raw data should be presented in figures whenever informative to do so (typically when N per group is less than 10)". In bar charts, authors should present each individual experiment as a data point.

*Reviewer #2 (Recommendations for the authors):*

Cecon et al. developed a series of tau biosensors based on NanoBiT complementation technology to monitor intramolecular and intermolecular tau interactions. Although the author made several variations of the biosensors in this study, these mutations and isoforms are presented in several other studies using other types of split-luciferase complementation to study spontaneous tau aggregation, phosphorylation and seeding. An interesting improvement is the use of split-luciferase to study intramolecular tau interaction. Using these luciferase biosensors, the authors show that phosphorylation of WT tau takes place when it is still bound to MTs and suggested that tau species comprising of K18 and full-length WT tau might represent an interesting new therapeutic target. Furthermore, the authors presented a proof-of-concept capability of their biosensors in high throughput screening drug discovery. While the overall concept is interesting, more data will be required to support the conclusion. I would recommend publication after substantial revision.

(1) It would be good to know what kind of tau-tau interaction is being formed in the biosensors, are they oligomers or fibrillar species? Perhaps ThS staining could provide more information on the type of interactions. Are there ways to differentiate between intramolecular and intermolecular interactions other than the luciferase signals – can the authors comment on the possibility of having intermolecular interaction in the intramolecular biosensors?

(2) For Figure 1D and 1E as well as for Figure 2A, what is the fixed amount of LgBit-K18(P301L) or WT tau used? Please make this clear to the readers. I supposed the SmBit will reach saturation at 1:1 ratio?

(3) In Figure 1G, can the authors explain why both monomeric K18 and aggregated K18 induced similar effect in the biosensor? In contrast, there is a difference when the biosensors are treated with mTau and aggTau.

(4) Can the authors use other measurements to show that tau aggregation is happening on the MT and that disruption of MT decreases tau aggregation? Please also include the immunofluorescence images of tubulin, tau and merged with treatment of colchicine. Lastly, it would be a good control to show that colchicine is not directly disrupting tau-tau interactions, perhaps through purified protein assays with treatment of colchicine. This will eliminate the reduced signals in Figure 2C coming from colchicine directly interfering with tau intermolecular aggregation.

(5) The intramolecular interaction of WT Tau has been shown previously but can the authors please comment on how tauRD could have intramolecular interaction? The repeat domains seem rigid and may not fold on each other. On the same note, it is an interesting concept to study the interaction between WT tau and K18(P301L). However, there is a huge length difference between WT tau and K18 tau. What is the linker length used between the NanoBiT fusion and tau? If the tau proteins are interacting at the repeat domains, is the linker length sufficiently long to allow interaction between WT tau and K18 tau to illustrate a signal?

(6) Can the authors please discuss if it is known that phosphorylation changes monomeric conformation? Can the authors please quantify the Western blots in Figure 2I and 2J, it appears the Fsk in Figure 2J is not inducing significant change of phosphorylation in tau. Colchicine has been suggested to dephosphorylate tau, could this be the reason to the reduced signals shown in the biosensor? Can the authors please show the Western blots of phosphorylation for Figure 2K in terms of the phosphorylation status upon colchicine treatment. The conclusion that phosphorylation is favored in MT-bound state needs further results to conclude.

(7) Regarding pg12, line 11, TauP301L is not showing up in the cytosols, hence it's difficult to conclude that it is existing as cytosolic tau, whereas K18(P301L) directly shows up in the cytosols. Did the authors try to create a K18 WT (without mutations) to see how it behaves as compared to the full-length WT? Can the authors please discuss the reason why the Tau(P301L) is not responding to the mK18 and aggTau which induced signals in K18(P301L)?

(8) In Figure 5B, why is there a basal signal of about 2000 with treatment of WT brain lysates whereas in Figure 4G, the WT brain lysates gave a lower signal?

(9) Can the authors please comment on how they will proceed to perform the HTS? Are they planning to screen thousands/millions of compounds with co-treatment of Tg brain lysates and the luciferase biosensors? As there is no direct testing of compounds with the luciferase biosensors in the absence of Tg brain lysates, it is difficult to tell if the compounds have a direct effect on the luciferase biosensors. The authors should also perform Z-factor calculations in the presence of the compounds and not just with the Tg brain lysates. It is of key importance to show that the biosensors can respond to the compounds in order to discover new drugs. What are the concentrations of compounds tested in Figure 5C and 5D? The authors should also calculate the IC50 of these compounds tested and compare their values with those in the literature. It appears that most compounds are only effective at very high concentration, which questions the validity of these results whether the effects are coming from other artifacts such as compound aggregation or other interfering properties.

(10) The authors stated that other techniques such as FRET has limitations in terms of sensitivity, laborious data analysis or in high-throughput screening (HTS) compatibility. While split luciferase has been suggested to have better sensitivity, it is unclear how this assay has been proven to be useful in HTS assay to identify novel compounds or how this assay is making the data analysis easier. The authors have used some compounds acting as positive controls to test that the small molecules could alter tau fibril formation and reduced the capability of fibrils in perturbing the NanoBiT biosensors. However, there is no real HTS conducted to prove that useful small molecules can be identified. In addition, no direct perturbation of the biosensor by the small molecules has been shown (authors added small molecules to break up fibrils which then change the NanoBiT signals). Furthermore, it is also not shown whether this assay is capable of screening for small molecules that target tau oligomers rather than the fibrils.

(11) Language and tone can be improved. For example, some words (e.g. Dementia) do not need to be capitalized and there are several spelling and grammar errors as well as more subtle tone can be used e.g. pg5 line 4 and 24, pg6 line 21, pg12, line5, pg15, line 17.

*Reviewer #3 (Recommendations for the authors):*

1) The punctuated pattern and the nuclear localization of K18(P301L) is not observed in other models using K18(P301S). If this is something that occurs due to the mutation it would be an interesting finding but would have to be confirmed by demonstrating that the same happens to the K18(P301SL) in the absence of the NanoBiT fragments.

2) The data related to the effect on phosphorylation needs further validation. Firstly the effects of forskolin are subtle based on the blots shown, other treatments that affect kinase activity should be used to confirm the findings. This should also be taken into account when employing the combination treatment with colchicine.

3) Page 16 line 19: the verb "implies" is used incorrectly.

4) Page 18 line 10: "sensibility" should be sensitivity.

5) Page 19 line 13: what does "cloned in front" mean? I'm assuming this would be either upstream or downstream depending on the sensor.

6) Page 19 line 16: the numbering in the description of the K18(P301L) is incorrect.

---

## [Author Response]

Essential revisions:Overall, while the reviewers see potential in this paper, substantial revisions are required. I urge you to go through the reviewers' comments thoroughly, as they bring up similar concerns about the need for better characterization of the biosensors, important controls, and toning down language to make the conclusions more reflective of the results.

We acknowledge the Editor’s summary of the essential revisions. The text has been revised to better reflect the results. New experiments addressing points 2 to 5 of your ‘Essential Revisions’ have been performed and are included in this revised version of the manuscript. Below is a point-by-point reply to the editors’ 6 points of the ‘Essential Revisions’ comments.

1) All three reviewers brought up concerns about the strong claims stated within the paper on how this assay is better than other currently available assays without providing any data to this effect. The authors must discuss other biosensors more thoroughly and either tone the language down substantially or repeat some of their experiments with other split-luciferase or fluorescent biosensors.

We revised the text, improved the Discussion on other biosensors and toned the language down. In addition, we performed new experiments, as detailed in the next points, to further characterize our biosensors and to compare their behavior with other sensors or with native tau proteins.

2) Can the authors use other measurements or assays to show that tau aggregation is occuring on the microtubule, and perhaps disrupt the microtubule network using another method such as nocodazole to show a decrease in tau aggregation? Please also include the immunofluorescence images of tubulin and tau both individually and merged with colchicine treatment. It would also be a good control to show that colchicine is not directly disrupting tau-tau interactions or affecting the phosphorylation state of tau.

We would like to clarify that we do not claim that tau aggregation is occurring on microtubules, as mentioned by reviewer 2, but that we rather think that our INTER-molecular sensor basal signal originates from molecular proximity of tau proteins individually attached to microtubules (MT). Our conclusion is also based on the fact that the tau-tau interaction is mutually exclusive with the tau-MT interaction as they both involve the same central repeat domain as shown in multiple studies (PMID: 2504257). Furthermore, colchicine treatment abolished the signal indicating that intact MTs are required for proximity of tau molecules. We confirmed such proximity with additional experiments using the proximity ligation assay (PLA) on the tau(P301L) interaction sensor. In the single-target PLA assay, the PLA signal is only observed if at least 2 molecules are in close proximity. PLA signals were observed along the MT network which was stained with anti-tubulin antibody confirming the close proximity of interacting sensor molecules at MTs. This data is now show in Figure 4F.

We also performed new experiments using nocodazole to disrupt microtubules and the results obtained are similar to those observed with colchicine (see new Figure 2 – supplementary figure S1B-D). Immunofluorescence images of tubulin and tau in cells treated with colchicine or nocodazole are also included now in Figures2D, I and 4C. Tau phosphorylation under colchicine treatment is also added in Figure 2M.

Lastly, to assess whether colchicine could directly disrupt the tau-tau interaction we produced recombinant full length WT tau and monitored the effect of colchicine on the heparin-induced tau-tau interaction with the thioflavin T assay. The addition of colchicine did not modify the ability of tau to self-interact indicating that colchicine is not directly disrupting the tau-tau interaction (see new Figure 2 – supplementary figure S1A and text page 9).

3) All of the reviewers agreed that there must be better characterization of the biosensors. The reviewers suggested that the authors should demonstrate that interaction and aggregation kinetics are not affected significantly compared to the native tau protein in vitro to support the physiological relevance of the claims related to inter- and intra-molecular interactions. Are there ways to differentiate between intramolecular and intermolecular interactions other than the luciferase signals? The colocalization data is not of high enough quality to support the claims regarding microtubule interactions, and a reviewer pointed out that there seems to be stronger colocalization with the intramolecular sensor than with the intermolecular one. Better quality images and co-localization analysis are needed to support these interpretations. A more extensive evaluation of the sensing capacity would be needed to establish sensitivity in a meaningful way, for instance with tau proteins for which concentration can be more appropriately estimated, e.g., recombinant tau and IP-purified extracts from mouse and human tissues, or a direct comparison with other methods. Finally, is it possible to further assay what kind of tau-tau interaction are being formed and recognized by the biosensors: are they oligomers or fibrillar species? Perhaps ThS staining could provide more information on the type of interactions.

In order to further characterize our sensor constructs, we decided to produce in *E. coli* the recombinant version of the LgBit-tagged Tau-P301L protein to compare its aggregation kinetics with those of the untagged Tau-P301L produced in parallel. Aggregation kinetics were measured in vitro with the ThT assay and turned out to be very similar, indicating that the 18kD LgBit fragment does not interfere with this process. Results are shown in Figure 4 – supplementary figure S1A (page 15).

Our aim in this project was to obtain a living cell system to monitor tau conformation and oligomerization. Apart from determining the static structure of the conformationally altered tau protein or its oligomers/fibrils in vitro, or by using single particles combined with fluorescence techniques in vitro, there is no dynamic method that we can think of, to monitor intramolecular or intermolecular changes of tau in living cells, except through the use of biosensors, e.g. generating a luminescence signal by reconstituting the luciferase activity as in our case here. The closest study to ours in terms of characterization of intramolecular tau conformational changes is the one from Di Primio et al. (PMID: 28713242), where they developed an intramolecular FRET-based tau sensor applied in live cells. They discriminated intra- versus intermolecular FRET signals by comparing the basal signal in cells expressing the intramolecular sensor or equal amounts of monolabeled tau (which would correspond to our inter-molecular sensor). They observed that at this equal expression condition the signal of the intramolecular sensor was significantly higher than that observed for the intermolecular FRET condition, and this higher signal is consistent with the predicted proximity of tau N- and C-terminal domains upon its hairpin three-dimensional conformation. We have thus performed a similar experiment with our biosensors by transfecting 250ng of the intramolecular LgBit-Tau-SmBit sensor or 125ng of each monotagged sensor construct (intermolecular sensor). Similarly, to Di Primio et al., we observe that, at equal level of expression, the basal luminescence signal from the intramolecular sensor is much higher (approximately 10 times) than that of the intermolecular sensor, suggesting thus that the large majority of the intramolecular signal indeed originates from the structural conformation of tau and not from aggregation between two separate tau molecules at the basal level. The data are shown in Figure 2F and is discussed in the text on page 09.

Immunofluorescence images of better quality and colocalization analysis are now shown in Figure 2D, 2I, 2J and 4C.

In order to further characterize the sensing capacity of our biosensor in a quantitative and comparable manner to other assays, we used recombinant K18 oligomers with a defined molarity as seeds. Concentrations as low as 0.3 nM of K18 oligomers were detectable with our sensor (see new Figure 1H). The FRET-based K18 sensor (Holmes et al. PMID: 25261551) reports a statistically significant signal increase with K18 aggregates as low as 326 fM. However, their sensor signal cannot be directly compared to ours. Indeed, while the FRET-based study exclusively selects positive cells (taking into account the % of FRET-positive cells and the FRET median fluorescence intensity), the detection method with our biosensor does not make any selection and takes into account all cells. Therefore, the major advantage of our sensor is that it does not require any algorithm for post-analysis of the data and the setup is much simpler compared to FACS analysis of FRET-positive cells. This is now discussed on pages 23.

To provide more information on the type of interactions detected with our sensor (oligomers or fibrillar species), we verified the co-occurrence of seed-induced fibrillar species, which are detected by ThS staining, with the localization of our sensors. As expected, ThS staining was only observed when cells were treated with Tg brain lysate seeds. Interestingly, it is not observed where the Tau(P301L) sensor is expressed, suggesting that the biosensor is in a molecular state not recognized by the ThS staining. A similar number of ThS-positive cells was observed in mock transfected cells, not expressing the biosensor, indicating that the ThS staining probably originates from the fibrillar species contained in the Tg brain lysate, and that the luminescent sensor signal (measured in parallel) corresponds rather to oligomeric species that are undetectable by ThS. These data are shown in Figure 4 —figure supplement S1D.

4) The reviewers agreed that the data related to the effect on phosphorylation needs further validation. It was noted that the effects of forskolin are subtle based on the blots shown, and perhaps other treatments that affect kinase activity could be used to confirm the findings. This should also be taken into account when employing the combination treatment with colchicine. The authors should provide blots of phosphorylation status upon colchicine treatment.

The effect of forskolin treatment on the phosphorylation of tau detected by the AT8 antibody reveals an increase in the level of tau phosphorylation by a factor of 2, even when we increased the Fsk concentration further (up to 40 µM). These results are in perfect agreement with the literature which also reveals a maximum fold change of 2 in western blot (PMID: 27995573). Longer incubation times (up to 48h) could possibly increase this signal further, however they affect cell proliferation and, consequently, the level of sensor expression, thus adding an important confounding factor to the interpretation of the sensor signal.

Therefore, to further confirm the impact of modulating tau phosphorylation on our sensor signal, we used okadaic acid, a phosphatase inhibitor described to increase tau phosphorylation. Similar to forskolin, a 2-fold increase in the signal of Tau WT intermolecular sensor was observed. The combined treatment with colchicine completely abolished the sensor signal, as in the case of the forskolin condition. These results are shown in Figure 2 – supplementary figure S1F.

To further document the phosphorylation state of tau upon colchicine treatment, new western blots are included in Figure 2M with quantification in Figure 2K, 2L and 2M, and show that tau phosphorylation correlates well with the increase in bioluminescence signal.

5) The reviewers were concerned about how the authors can use the biosensors directly to screen for small molecules that affect tau aggregation with their current biosensor as a read-out. The authors should perform Z-factor calculations on the small molecules as positive controls with direct relevance to the biosensors and not indirectly through affecting the Tg brain lysates. The authors should also calculate the IC50 of the compounds tested and compare their values with those in the literature, as well as clearly state the concentrations they are using. There was also a recommendation that the authors test methylene blue, which is in phase 3 for its anti-tau aggregation properties.

We apologize to reviewer 2 if the first version of the manuscript was not clear on the screening aspect. The advantage of our biosensor is that it does not generate high signal if not induced by pathological seeds. We therefore propose to use the luciferase biosensor in the presence of Tg brain lysates (together as a system), and *not* the biosensor alone. Thus, this system (sensor+seeds) can be used to screen for drug candidates with *anti-seeding properties.* When not induced by pathological seeds, the sensors alone are not expected to respond to potential inhibitory compounds of Tau aggregation. Furthermore, as a technical control, we also verified that positive hits did not affect by themselves the luciferase activity (Figure 5 – supplementary figure S1B).

Concerning the calculation of the Z-factor, we previously used the term “positive control” to refer to the positive signal controls induced by seeds, and not to a positive control of a hit compound. To avoid any misunderstandings, we have now modified the text by replacing “positive control” and “negative control” by “high reference control” and “low reference control”, respectively. Hence, our high reference control corresponds to the sensor signal in the presence of Tg brain lysates, while the low reference control is the sensor signal in the absence of pathological seeds (here we used the WT brain lysates as a vehicle control). Consequently, the calculated Z-factor indicates that the assay displays a wide window of separation between the high and low references with low variability, suggesting thus a good performance for screening assay application.

The IC50 of the compounds is now included in the text (page 17) and correspond well to the values previously reported in the literature. This aspect is now included in the Discussion (page 23).

The concentrations of compounds used in Figure 5C-D are now stated in the figure legend.

Instead of including methylene blue in our study, we used LMTMeSO4 which is a second-generation compound derived from methylene blue with an improved profile that has replaced methylene blue in clinical trial studies. Methylene blue was discontinued at phase II and LMTMeSO4 is currently in a phase 3 clinical trial (Harrington et al., 2015; VandeVrede et al., 2020). This is now included in the Discussion (page 24).

Reviewer #1 (Recommendations for the authors):Some recommendations and suggestions are provided below to improve the quality of this very interesting work.1) The suitability of tau biosensors for high-throughput screening needs to be further developed as experiments were only performed in 96-well plates. No scaling up in 384- or 1536-well plates was attempted. Moreover, the transfection of the biosensors could be tedious and the generation of a stable cell line could have been envisaged.

We acknowledge the reviewer’ suggestions. We have performed the aggregation assay with the K18(P301L) intermolecular sensor in 384-well plate and obtained similar results to those previously observed in the 96-well plate format. This result is now shown in Figure 3 – supplementary figure 1C. The generation of a stable cell line is indeed envisaged and in progress and we intend to provide the scientific community with such a cell line, once identified the optimal absolute and relative expression of both LgBit- and SmBit-tau constructs.

2) The models used are basic (HEK-293T cells associated with tau aggregates, Aβ oligomers or mouse brain lysates). We might regret that no other cellular model of neurodegenerative disease has been considered, such as the lentiviral expression of biosensors in an IPS cell line derived from a patient carrying a tau mutation. The number of drugs tested is also quite low. Methylene blue, which is tested in phase 3 for its anti-tau aggregation properties, could have been tested.

To test our sensors in a more neuron-like cellular context we chose the human neuroblastoma-derived SH-SY5Y cell line and were able to replicate the forskolin effect observed in HEK293 cells with the conformational full-length WT tau sensor (Figure 2 —figure supplement 1G).

The question related to Methylene blue has been addressed in our answer to “Essential Revisions (for the authors), point 5”.

ResultsPage 6, line 23: The nuclear localization of the biosensor is quite substantial and must be commented on.

We agree with the reviewer and we comment this more in detail in the revised version (page 20).

Page 7, lines 1-7: A Western Blot presenting the aggregated products (K18, Full-length tau, Aßo) used in comparison with the non-aggregated forms could be presented. In addition, please provide indications regarding the quantity of proteins and Aßo that were used in the experiments.

Successful aggregation of K18, tau and Aß has been accessed by thioflavin T assay and these data are now show in Figure 1—figure supplement 1A. Information on the quantities is now added in Methods and all figure legends.

Figures 2C, 2D and 4D, 4E and corresponding text: Visualizing by microscopy the effect of colchicine treatments could reinforce the demonstration.

This aspect has been addressed in our answer to “Essential Revisions (for the authors), point 2 ». (Figures 2D, 2I and 4C).

Figure 2E: Please indicate error bars on both sides to be consistent with Figure 2A.

Corrected.

Page 9, line 2: Please correct P301L.

Corrected.

Page 9, line 13: Please briefly explain the mode of action of forskolin (PKA activator) and edit line 16 as it sounds like forskolin itself is responsible for tau phosphorylation.

Corrected (page 10).

Figures 2I, 2J and 4A, 4B, and corresponding text: Please provide quantification of the blots showing phosphorylation of tau biosensors.

This aspect has been addressed in our answer to “Essential Revisions (for the authors), point 2” (Figures 2K, 2L, 2M and 4A, 4B).

Page 9, lines 22-23: Co-treatment with colchicine followed by forskolin was performed on interaction WT-Tau sensors. What is the effect on the conformation WT-Tau sensors?

This aspect has been addressed and is now shown in Figure 2 —figure supplement 1C,E.

Page 9, line 25: "to monitor the behavior of tau in a physiological cellular context" and Page 17 line 1 "physiological context": In my opinion, this is not a real "physiological" cellular context as the biosensors have been overexpressed in the cells. I would rather use the term "living cell" This also raise the point of the amount of tau biosensor versus the level of endogenous tau. Did the authors verify this point using tau antibodies recognizing both endogenous tau and biosensors?

We have modified the text as suggested. The expression level of the biosensors has not been directly compared to neuronal endogenous tau. We suspect that the expression level is not excessively high based on the immunofluorescence results in both, HEK cells (Figure 2D,I) and in the neuronal cell line SH-SY-5Y (Figure 2 —figure supplement S1G), where the tau sensor is not artificially mislocalized but rather decorates the microtubules, as commonly seen in neuronal primary cultures reported in the literature (i.e. https://doi.org/10.1091/mbc.E19-03-0183).

Figure 3 supplement 1 and 2: I would merge these two figures, and use more conventionally the terminology of Supplementary Figures 1 to 4, unless it is specific the Journal.

Figure 3 supplements 1 and 2 have been merged. The terminology of Supplementary Figures is used according to the specific rules of the Journal.

Figure 4H left panel and corresponding text: How does the authors explain that not all cells expressing the biosensors show MC1 staining? Could this be based on an induction threshold for MC1 staining? Regarding AT8 staining, it would be preferable to present an image showing several cells.

The MC-1 antibody recognizes a specific pathological conformation of tau (PMID: 9130141) and it is thus possible that not all biosensor molecules reached this conformation in the experimental conditions used (*i.e.* 24h treatment with seeds). It is difficult to judge the expected percentage of positive cells as only few reports in the literature attempted to probe the co-localization of MC-1 staining with other tag-targeted antibodies, and in the few cases where this is reported, only one positive co-localization cell is shown (example: PMID 17008320). In addition, the reviewer is right and it is indeed very plausible that a certain threshold on the number of tau molecules with MC-1-reactive conformation is required to be detectable by conventional confocal microscopes.

We have replaced all the images of Figure 4I by images with more sensor-positive cells, as suggested.

MethodsPage 20, line 5: Contrary to what is indicated here, the amount of each construct is not always indicated. For reproducibility purpose, the amount of cDNA transfected should be indicated in the legend of each figure panel, along with the number of cells used.

Corrected.

Page 20, line 17: Please replace "Phosphate de sodium" by "Sodium phosphate".

Corrected.

Page 20, line 18: Please replace "acétate d'ammonium" by "ammonium acetate".

Corrected.

Page 21, line 6: Please indicate how many mice were used. In the case of pooled extracts, how many different mice were mixed in the pools?

Corrected.

Page 21, line 20: The "micro" (µ) signs have been replaced by u signs.

Corrected.

Page 22, line 2: Please mention here that an acquisition time of 10 minutes was used for all graphs, unless otherwise stated.

Corrected.

Page 23, line 5: Please define TBST.

Corrected.

Page 24, line 5: The authors indicate that they have calculated the Z-prime (Z', calculated from the medians) but they indicate the formula of the Z-factor (calculated from the means). Please clarify.

We used the means to calculate the Z-factor. We have corrected this in the text (Z’ has been replaced by Z-factor; pages 17 and 34).

FiguresAccording to eLife policies, "Raw data should be presented in figures whenever informative to do so (typically when N per group is less than 10)". In bar charts, authors should present each individual experiment as a data point.

Raw data is presented whenever informative. Pooled normalized data are presented when statistical analysis are performed. Data needs to be normalized to a reference within each experiment (reference being mock treated cells, or treatment with WT brain lysates or vehicle) before being pooled, because of the variations inherent to each experiment (level of expression, number of cells, etc). We have modified all the graphs and we now display each individual experiment data point with the mean and S.E.M.

Reviewer #2 (Recommendations for the authors):Cecon et al. developed a series of tau biosensors based on NanoBiT complementation technology to monitor intramolecular and intermolecular tau interactions. Although the author made several variations of the biosensors in this study, these mutations and isoforms are presented in several other studies using other types of split-luciferase complementation to study spontaneous tau aggregation, phosphorylation and seeding. An interesting improvement is the use of split-luciferase to study intramolecular tau interaction. Using these luciferase biosensors, the authors show that phosphorylation of WT tau takes place when it is still bound to MTs and suggested that tau species comprising of K18 and full-length WT tau might represent an interesting new therapeutic target. Furthermore, the authors presented a proof-of-concept capability of their biosensors in high throughput screening drug discovery. While the overall concept is interesting, more data will be required to support the conclusion. I would recommend publication after substantial revision.(1) It would be good to know what kind of tau-tau interaction is being formed in the biosensors, are they oligomers or fibrillar species? Perhaps ThS staining could provide more information on the type of interactions. Are there ways to differentiate between intramolecular and intermolecular interactions other than the luciferase signals – can the authors comment on the possibility of having intermolecular interaction in the intramolecular biosensors?

This aspect has been addressed in our answer to “Essential Revisions (for the authors), point 3”.

(2) For Figure 1D and 1E as well as for Figure 2A, what is the fixed amount of LgBit-K18(P301L) or WT tau used? Please make this clear to the readers. I supposed the SmBit will reach saturation at 1:1 ratio?

Amounts of transfected constructs are indicated in the figure legends. Given that we study a state of aggregation here, we expect that, upon induction with seeds, the signal will reach saturation when all LgBit molecules have complemented with SmBit molecules, either in a 1:1 ratio or beyond with an excess of SmBit over LgBit.

(3) In Figure 1G, can the authors explain why both monomeric K18 and aggregated K18 induced similar effect in the biosensor? In contrast, there is a difference when the biosensors are treated with mTau and aggTau.

The K18 recombinant protein is more prone to form aggregates than full-length WT tau. Although we controlled the initial preparation of the recombinant proteins to ensure their initial state as monomers or aggregates, we cannot exclude that the 24h incubation time with the cells, was sufficient for the monomeric recombinant K18 to aggregate and, thus, to induce the aggregation of the sensor. The presence of the K18 sensor itself could also have contributed to initiate the nucleation of the recombinant K18. In contrast to K18, full-length WT tau does not readily form aggregates and, thus, the monomeric form should be stable over time in culture as observed in our experiments.

(4) Can the authors use other measurements to show that tau aggregation is happening on the MT and that disruption of MT decreases tau aggregation? Please also include the immunofluorescence images of tubulin, tau and merged with treatment of colchicine. Lastly, it would be a good control to show that colchicine is not directly disrupting tau-tau interactions, perhaps through purified protein assays with treatment of colchicine. This will eliminate the reduced signals in Figure 2C coming from colchicine directly interfering with tau intermolecular aggregation.

This aspect has been addressed in our answer to “Essential Revisions (for the authors), point 2”.

(5) The intramolecular interaction of WT Tau has been shown previously but can the authors please comment on how tauRD could have intramolecular interaction? The repeat domains seem rigid and may not fold on each other. On the same note, it is an interesting concept to study the interaction between WT tau and K18(P301L). However, there is a huge length difference between WT tau and K18 tau. What is the linker length used between the NanoBiT fusion and tau? If the tau proteins are interacting at the repeat domains, is the linker length sufficiently long to allow interaction between WT tau and K18 tau to illustrate a signal?

The basal luminescence signal of the intramolecular K18(P301L) sensor is detectable despite the intrinsic rigidity of this domain. The probability of complementation of the LgBiT and the SmBiT fragments is higher within the K18 molecule due to the short size of this domain (40 amino acids) and the flexible linkers: one between LgBit and K18 of 25 amino-acids and one between K18 and SmBit of 30 amino-acids.

Consistent with the notion that the K18 domain is rigid, the majority of the experimental conditions tested was unable to induce a change of the basal signal of the K18 intramolecular sensor, implying that no detectable conformational change occurred. The only exception was with the recombinant aggregated K18 seeds, which partially diminished the signal (Figure 3 – supplementary figure S1D). We favor the hypothesis that the decreased of the intramolecular signal of the K18 sensor, induced by treatment with K18 aggregates, could be the result of an incorporation of the K18 sensor molecules into the aggregated K18 seeds, which could then indirectly interfere with the proximity of intramolecular C-and N-terminal domains.

Regarding the interaction between full-length WT tau and K18 tau, the intermolecular sensors have the same linker composition and length. According to the predicted paper-clip structure of tau full-length, the N-terminal domain folds back towards the central 4-repeat domain, where the interaction of full-length WT tau with K18 should occur. This would allow close proximity between the N-terminal ends of both proteins and, thus, the Nluc complementation.

(6) Can the authors please discuss if it is known that phosphorylation changes monomeric conformation? Can the authors please quantify the Western blots in Figure 2I and 2J, it appears the Fsk in Figure 2J is not inducing significant change of phosphorylation in tau. Colchicine has been suggested to dephosphorylate tau, could this be the reason to the reduced signals shown in the biosensor? Can the authors please show the Western blots of phosphorylation for Figure 2K in terms of the phosphorylation status upon colchicine treatment. The conclusion that phosphorylation is favored in MT-bound state needs further results to conclude.

This aspect has been addressed in our answer to “Essential Revisions (for the authors), point 2”.

A conformational change of monomeric tau has previously been suggested by the use of intramolecular FRET-based sensors including tau mutated at specific phosphorylation sites (Di Primio et al., 2017 PMID: 28713242), and more recently in cells expressing another FRET sensor and treated with okadaic acid (Rudenko et al., 2019 PMID: 31524157). This is now in the Discussion (page 19).

The Western blot of phosphorylated tau is now shown (Figure 2M). Indeed, we observe less phosphorylation in the cells treated with colchicine. Our hypothesis is that MT destabilization by colchicine leads to dissociation of tau from microtubules (due to competitive binding, previously shown PMID: 284377) which assumes a conformation that modifies the accessibility to kinases and/or phosphatases. Our new data with okadaic acid (Figure 2 – supplementary Figure S1F) showing the same response profile as forskolin in the presence of colchicine further suggest that the accumulation of tau phosphorylation requires intact MT.

(7) Regarding pg12, line 11, TauP301L is not showing up in the cytosols, hence it's difficult to conclude that it is existing as cytosolic tau, whereas K18(P301L) directly shows up in the cytosols. Did the authors try to create a K18 WT (without mutations) to see how it behaves as compared to the full-length WT? Can the authors please discuss the reason why the Tau(P301L) is not responding to the mK18 and aggTau which induced signals in K18(P301L)?

The cytoplasmic and MT distribution of the Tau(P301L) has been confirmed in new experiments using Proximity Ligation Assay (PLA) (new Figure 4F). We did not create a K18 WT sensor (without mutations) because a previous study using YFP-tagged sensors showed a similar aggregation property of K18 WT compared to K18 with pro-aggregation mutations (Sanders et al., 2014 PMCID: PMC4171396). We therefore expect that a K18 WT sensor would behave more like our K18(P301L) sensor and not like our full-length tau WT sensor, which is resistant to aggregation.

By comparing the overall results of the Tau(P301L) and the K18(P301L) sensors we find that the response amplitude is lower in the Tau(P301L). The lack of response of the Tau(P301L) sensor to mK18 and aggTau could therefore be due to the lower sensitivity of this sensor compared to K18(P301L). Another possibility is that the Tau(P301L) is less exposed to seeds compared to K18 where the aggregation-prone 4R domain of Tau(P301L) could be more hidden. Finally, we also observe higher basal signal in the Tau(P301L) sensor, due to the proximity of MT-bound Tau(P301L) sensor molecules, compared to K18(P301L), which could thus mask an aggregation of small amplitude.

(8) In Figure 5B, why is there a basal signal of about 2000 with treatment of WT brain lysates whereas in Figure 4G, the WT brain lysates gave a lower signal?

This difference is due to the fact that the data in Figure 5B is the raw luminescence signal, while in the previous Figure 4G (4H in the revised version) the data is normalized to % of the basal (i.e. the vehicle treated cells).

(9) Can the authors please comment on how they will proceed to perform the HTS? Are they planning to screen thousands/millions of compounds with co-treatment of Tg brain lysates and the luciferase biosensors? As there is no direct testing of compounds with the luciferase biosensors in the absence of Tg brain lysates, it is difficult to tell if the compounds have a direct effect on the luciferase biosensors. The authors should also perform Z-factor calculations in the presence of the compounds and not just with the Tg brain lysates. It is of key importance to show that the biosensors can respond to the compounds in order to discover new drugs. What are the concentrations of compounds tested in Figure 5C and 5D? The authors should also calculate the IC50 of these compounds tested and compare their values with those in the literature. It appears that most compounds are only effective at very high concentration, which questions the validity of these results whether the effects are coming from other artifacts such as compound aggregation or other interfering properties.

This aspect has been addressed in our answer to “Essential Revisions (for the authors), point 5”.

(10) The authors stated that other techniques such as FRET has limitations in terms of sensitivity, laborious data analysis or in high-throughput screening (HTS) compatibility. While split luciferase has been suggested to have better sensitivity, it is unclear how this assay has been proven to be useful in HTS assay to identify novel compounds or how this assay is making the data analysis easier. The authors have used some compounds acting as positive controls to test that the small molecules could alter tau fibril formation and reduced the capability of fibrils in perturbing the NanoBiT biosensors. However, there is no real HTS conducted to prove that useful small molecules can be identified. In addition, no direct perturbation of the biosensor by the small molecules has been shown (authors added small molecules to break up fibrils which then change the NanoBiT signals). Furthermore, it is also not shown whether this assay is capable of screening for small molecules that target tau oligomers rather than the fibrils.

This aspect has been addressed in our answer to “Essential Revisions (for the authors), point 5”.

We apologize if our point was not clear in the first version of the manuscript. Our intention in the current study was not to perform a HTS screen but to provide proof of concept that HTS screening is possible with our biosensor: (i) by determining the Z-factor, and (ii) by testing the response of our sensor to small molecules that have been reported to exhibit anti-aggregation activity. Furthermore, the interest of our biosensor lies in the simplicity of the experimental methodology and the simple analysis of the data (suitable to HTS): measurement of the luciferase signal in 96- or 384-well plates on replicate measurement and simple comparison of the readily available signals on treated and vehicle-treated cells.

The reviewer is perfectly right and it is indeed true that we cannot distinguish whether the compounds target the fibrils from the brain lysates, or the sensor molecule itself. However, our aim here is to obtain a HTS cell-based *seeding assay* where positive hits would prevent tau propagation regardless whether their target is the sensor (monomeric tau) or the pre-existent fibrils (the cause of the sensor response). Further research into the specific mechanism of action and the specificity of the positive hits coming out from a primary screen will in any case always be necessary, whatever the screening assay used.

The question of whether the identified molecules interfere with oligomer or fibril formation (or both) is indeed relevant. As previous methods were unable to distinguish between interference with small oligomers formation, there is no available data on this point. We hope that our new sensor will allow to start addressing this issue now.

More generally, the split Nanoluc assay has already been successfully used in screening campaigns to identify small molecular weight PPI inhibitors (PMID: 26569370; PMID: 32587898), demonstrating the suitability of this technology for HTS.

(11) Language and tone can be improved. For example, some words (e.g. Dementia) do not need to be capitalized and there are several spelling and grammar errors as well as more subtle tone can be used e.g. pg5 line 4 and 24, pg6 line 21, pg12, line5, pg15, line 17.

The word ‘dementia’ is now lowercase. The text has been modified at the indicated locations and the reminder of the text has been checked for spelling and grammar errors and tone.

Reviewer #3 (Recommendations for the authors):1) The punctuated pattern and the nuclear localization of K18(P301L) is not observed in other models using K18(P301S). If this is something that occurs due to the mutation it would be an interesting finding but would have to be confirmed by demonstrating that the same happens to the K18(P301SL) in the absence of the NanoBiT fragments.

A recent study from Lester et al. (2021 PMCID: PMC8141031) shows that in both mouse and cell models of tau pathologies, K18 with the P301S mutation forms cytosolic tangles and nuclear puncta. A previous report on a fluorescence-based aggregation sensor including K18-YFP or the double-mutated K18(P301L/V337M)-YFP also showed that, after treatment with K18 fibrils, nuclear localization of the YFP sensor aggregates was observed (Sanders et al., 2014 PMCID: PMC4171396). It appears therefore that the presence of K18 in the cell nucleus is independent of the P301L/S mutation. As the cell distribution of these YFP sensors prior to the treatment with seeds to induce aggregation is not shown, it is difficult to discriminate if the nuclear localization of these YFP sensors is restricted to their aggregated form. Our own data, however, support the existence of nuclear K18(P301L) in the absence of aggregation. Of note, nuclear tau aggregates were also observed in a seeding model of tau expressing full-length P301S 0N4R tau-YFP in H4 neuroglioma cells (Lester et al., 2021 PMCID: PMC8141031), while our full-length P301L 2N4R tau sensor did not reveal prominent nuclear localization either before or after seeding. This difference further argues against a major impact of the P301L/S mutation on the determination of tau nuclear localization.

This discussion is now added to the text (page 20).

2) The data related to the effect on phosphorylation needs further validation. Firstly the effects of forskolin are subtle based on the blots shown, other treatments that affect kinase activity should be used to confirm the findings. This should also be taken into account when employing the combination treatment with colchicine.

This aspect has been addressed in our answer to “Essential Revisions (for the authors), point 4” and new experiments have been performed using the phosphatase inhibitor okadaic acid.

3) Page 16 line 19: the verb "implies" is used incorrectly.

Corrected.

4) Page 18 line 10: "sensibility" should be sensitivity.

Corrected.

5) Page 19 line 13: what does "cloned in front" mean? I'm assuming this would be either upstream or downstream depending on the sensor.

Corrected.

6) Page 19 line 16: the numbering in the description of the K18(P301L) is incorrect.

Corrected.